SciPost Physics

Submission

# Impurities in systems of noninteracting trapped fermions

David S. Dean[1], Pierre Le Doussal[2], Satya N. Majumdar[3], Grégory Schehr[3*]

**1** Univ. Bordeaux and CNRS, Laboratoire Ondes et Matière d'Aquitaine (LOMA),
UMR 5798, F-33400 Talence, France
**2** CNRS-Laboratoire de Physique Théorique de l'Ecole Normale Supérieure, 24 rue
Lhomond, 75231 Paris Cedex, France
**3** Université Paris-Saclay, CNRS, LPTMS, 91405, Orsay, France
* gregory.schehr@u-psud.fr

January 9, 2021

## Abstract

We study the properties of spin-less non-interacting fermions trapped in a confining potential but in the presence of one or more impurities which are modelled by delta function potentials. We use a method based on the single particle Green's function. For a single impurity placed in the bulk, we compute the density of the Fermi gas near the impurity. Our results, in addition to recovering the Friedel oscillations at large distance from the impurity, allow the exact computation of the density at short distances. We also show how the density of the Fermi gas is modified when the impurity is placed near the edge of the trap in the region where the unperturbed system is described by the Airy gas. Our method also allows us to compute the effective potential felt by the impurity both in the bulk and at the edge. In the bulk this effective potential is shown to be a universal function only of the local Fermi wave vector, or equivalently of the local fermion density. When the impurity is placed near the edge of the Fermi gas, the effective potential can be expressed in terms of Airy functions. For an attractive impurity placed far outside the support of the fermion density, we show that an interesting transition occurs where a single fermion is pulled out of the Fermi sea and forms a bound state with the impurity. This is a quantum analogue of the well-known Baik-Ben Arous-Péché (BBP) transition, known in the theory of spiked random matrices. The density at the location of the impurity plays the role of an order parameter. We also consider the case of two impurities in the bulk and compute exactly the effective force between them mediated by the background Fermi gas.

# 1   Introduction

Noninteracting fermions in a confining trap is a topic of much current interest, especially in the context of cold atoms. In the presence of a trap, the density of the Fermi gas in the ground state is confined in a finite region of space. Indeed, the density vanishes outside a finite interval in one-dimension. Inside this interval, usually referred to as the "bulk", the fermion density can be estimated, for a large number of fermions $N$, using a semi-classical approximation, or equivalently the so-called local density approximation (LDA) [1–3]. Near the edge where the density vanishes, the quantum fluctuations play a dominant role and the local properties of the fermions are very different from that of the bulk [4–7]. This edge region is called the "Airy gas" because the Airy functions play an important role in describing the quantum correlations. It is well known that the LDA is a very good approximation when the confining potential is smooth. However, if this potential has singularities, such as a step or delta-function, this method fails. In a recent paper [8], we have examined the effect of a step potential, using exact methods based on the determinantal properties of the Fermi gas.

Here, we consider instead the case when the smooth confining potential is modulated by introducing one or more delta functions. This situation naturally arises when one introduces one or more immobile impurities in the Fermi gas, where the impurities are modelled by delta-function potentials (attractive or repulsive). In the absence of the trap, i.e., for a free Fermi gas, the effects of such impurities have been well studied in the literature. For example, when a single impurity is introduced in the Fermi gas, the density of the Fermi gas is modulated near the impurity. At distances far from the impurity, the density exhibits decaying oscillations, famously known as "Friedel oscillations" [9–11]. How this density gets modulated close to the impurity has been studied in [12] in one dimension for delta function potentials. In addition, the effective Casimir interaction between two

impurities (mediated by the background Fermi gas) has also been studied [13,14], however, again, the results obtained are only valid at large distances. Note that similar questions have also been studied for bosonic systems, with and without interactions (see e.g. [15,16]).

In this paper we employ a method based on the single particle Green's function that allows us to obtain exact results for the trapped Fermi gas. We first consider the case when a single impurity is added to the trapped Fermi gas in three different locations: (i) in the bulk, (ii) near the edge and (iii) outside the edge. In the bulk (case (i)), where the trapped Fermi gas behaves locally as a free Fermi gas, we obtain the explicit form of the density near the impurity at all scales, not necessarily large. At large distances, we recover Friedel oscillations, our short distance results agree with those of [12] when the impurity is repulsive but we show that a correction is needed for attractive impurities . However in cases (ii) and (iii), the presence of the trap considerably modifies the bulk results and we obtain new results for the density close to the impurity. In addition, in all the three cases, we compute the effective potential felt by the impurity due to the background Fermi gas. In case (iii), for an attractive impurity, we show that an interesting transition occurs where a single fermion is pulled out of the Fermi sea and forms a bound state with the impurity. This is a quantum analogue of the classical Baik-Ben Arous-Péché (BBP) transition [17,18], known in the theory of spiked random matrices, where an eigenvalue detaches from the bounded support of the eigenvalues, due to a rank-one perturbation. This rank-one perturbation is the analogue of the delta-function potential induced by the impurity in the Fermi gas and the eigenvalue is the analogue of the fermion position. We then go beyond the case of a single impurity and study the effects of adding two impurities. In this case we compute the effective Casimir-like interaction between the two impurities mediated via the trapped Fermi gas. For two impurities, we restrict our analysis to case (i) where both the impurities are placed in the bulk. In this case, we obtain exact results for this effective interaction at *arbitrary* separation between the impurities. At large distances, our results are in agreement with the one found in Refs. [13,14] obtained by a different method.

We restrict ourselves here to the case of *immobile* impurities. The case of mobile impurities has been extensively studied for fermionic systems both theoretically [19–25] and experimentally [26]. The set up of immobile impurities that we study in this paper is more difficult to access experimentally, however it has been suggested that impurities could be introduced by superimposing an optical lattice on an overall trapping potential [27].

## 2 Model and summary of main results

### 2.1 The model

We consider a gas of identical spin-less fermions of mass $m$ in a trap generated by a potential $V(x)$ at zero temperature. We then add $n$ delta-impurities of strengths $g_i$. The single particle Hamiltonian is then given by

$$H = H_0 + \Delta H \quad , \qquad H_0 = -\frac{\hbar^2}{2m}\frac{\partial^2}{\partial x^2} + V(x) \quad \text{and} \quad \Delta H = \sum_{i=1}^{n} g_i \delta(x - x_i) \ , \quad (1)$$

where $H_0$ is the Hamiltonian associated with the trap and $\Delta H$ corresponds to the impurities. The eigenfunctions and eigenvalues of $H_0$ are denoted by $\psi_k^0(x)$ and $\epsilon_k^0$. Similarly, $\psi_k(x)$ and $\epsilon_k$ denote the eigenfunctions and eigenvalues of $H$. Consider first the case with no impurity (i.e. $g_i = 0$ for all $i = 1, \cdots, n$). At zero temperature, the system is in the many-body ground-state such that all single-particle states of $H_0$ below the Fermi level,

denoted by $\mu$, are occupied, each by a single fermion. The ground-state energy is given by

$$E_0(\mu) = \sum_k \theta(\mu - \epsilon_k^0)\epsilon_k^0 \; , \tag{2}$$

where $\theta(x)$ is the Heaviside function where one usually uses the definition $\theta(0) = 1$. We now switch on the $g_i$'s, i.e., we introduce the impurities in the system. The single-particle Hamiltonian then changes from $H_0$ to $H$. This will change the single-particle eigenfunctions and eigenvalues. Consequently the ground state energy will also change. Here we work in the grand-canonical ensemble where the Fermi level $\mu$ remains fixed, while the number of fermions is not fixed and the system is in contact with a reservoir of particles. Then the new ground state energy, in the presence of the impurities, is given by

$$E(\mu) = \sum_k \theta(\mu - \epsilon_k)\epsilon_k \; . \tag{3}$$

Similarly one can define the number of particles $N_0(\mu)$ and $N(\mu)$ below the Fermi level $\mu$ as

$$N_0(\mu) = \sum_k \theta(\mu - \epsilon_k^0) \quad , \quad N(\mu) = \sum_k \theta(\mu - \epsilon_k) \; . \tag{4}$$

Since we are working in the grand-canonical setting, $N_0(\mu)$ and $N(\mu)$ can be different and the quantity which will play a crucial role is the grand-potential at zero temperature

$$\Omega(\mu) = E(\mu) - \mu N(\mu) \; . \tag{5}$$

At zero temperature, thanks to the Wick's theorem, all the correlation functions are given by determinants constructed from the so-called *kernel*, which reads for $H_0$ and $H$ respectively [5]

$$K_{0\mu}(x,y) = \sum_k \theta(\mu - \epsilon_k^0)\psi_k^{0*}(x)\psi_k^0(y) \quad , \quad K_\mu(x,y) = \sum_k \theta(\mu - \epsilon_k)\psi_k^*(x)\psi_k(y) \; . \tag{6}$$

Setting $x = y$ in the Eq. (6) we find the fermion density

$$\rho_{0\mu}(x) = K_{0\mu}(x,x) = \sum_k \theta(\mu - \epsilon_k^0)|\psi_k^0(x)|^2 \quad , \quad \rho_\mu(x) = K_\mu(x,x) = \sum_k \theta(\mu - \epsilon_k)|\psi_k(x)|^2 \; . \tag{7}$$

## 2.2 Outline and main results

In Section 3 A, we first recall the method of the Green's function introduced in [8] and present in Section 3 B the explicit expressions for the Green's function in the absence of impurities. This is then used as the building block to obtain an exact expression for the Green's function in the presence of the impurities in Section 3 C. In Section 4 we examine the effect of impurities in the bulk of the system. In the absence of the impurities, the density in the bulk is given by

$$\rho_{0\mu}(x) = K_{0\mu}(x,x) \simeq \frac{k_F(x)}{\pi} \qquad \text{where} \qquad k_F(x) = \frac{1}{\hbar}\sqrt{2m(\mu - V(x))} \; . \tag{8}$$

Here $k_F(x)$ is just the local Fermi wave vector. The density vanishes at the edge $x_e$ where $V(x_e) = \mu$. The bulk of the system is thus defined such that $V(x) \ll \mu$ (see below in Eq. (29) for a more precise definition). We now add a single impurity in the bulk of the

system, say at $x = 0$, and investigate how the kernel and the density change near the impurity. We reparametrize the impurity strength in terms of an inverse length scale $\lambda$ defined as

$$g = \frac{\lambda \hbar^2}{m} . \tag{9}$$

We show that the change in the kernel upon adding this impurity is given by

$$\Delta K_\mu(x, y) = K_\mu(x, y) - K_{0\mu}(x, y) = \frac{\lambda \exp(\lambda(|x| + |y|))}{\pi} \mathrm{Im}\ \mathrm{E}_1(\lambda + ik_F)(|x| + |y|)] , \tag{10}$$

where Im denotes the imaginary part and $\mathrm{E}_1(z) = \int_z^\infty dt\, e^{-t}/t$ denotes the exponential integral [28]. Here $k_F = k_F(0)$ is the Fermi wave vector at the impurity position. Putting $x = y$ in (10) we obtain the change in the density due to the impurity (see also Figs. 4 and 5)

$$\Delta \rho_\mu(x) = K_\mu(x, x) - K_{0\mu}(x, x) = \frac{\lambda \exp(2\lambda|x|)}{\pi} \mathrm{Im}\ \mathrm{E}_1(2(\lambda + ik_F)|x|) . \tag{11}$$

We show how this formula recovers the density change computed in [12] for a free Fermi gas when the impurity $\lambda$ is repulsive. The result given here and in [12] is exact for a homogeneous free fermion system at all impurity strengths and distances and we note that formulas given for these *Friedel oscillations* [9–11] are often given in the regime of linear response or at long distances. However our formula in (11) holds for any $x$ and $\lambda$. In particular, we obtain an explicit formula for the density at the position of the impurity

$$\rho_\mu(0) = \frac{k_F}{\pi} - \frac{\lambda}{\pi} \arg(ik_F + \lambda) = \frac{k_F}{\pi} - \frac{\lambda}{2} + \frac{\lambda}{\pi} \tan^{-1}\left(\frac{\lambda}{k_F}\right) . \tag{12}$$

We then compute the effective potential $V_{\mathrm{eff}}(x_1)$ felt by the impurity, where $x_1$ denotes the position of the impurity in the bulk, due to its interaction with the Fermi gas. This is obtained by computing the change in the ground state energy of the many-body system due to the addition of an impurity. This effective potential can be expressed in the scaling form

$$V_{\mathrm{eff}}(x_1) = \Omega(\mu) - \Omega_0(\mu) = \frac{\hbar^2 \lambda^2}{2\pi m} W\left(\frac{k_F(x_1)}{\lambda}\right) , \tag{13}$$

where $\Omega_0(\mu) = E_0(\mu) - \mu N_0(\mu)$ and the scaling function is given by

$$W(\gamma) = (\gamma^2 + 1) \tan^{-1}\left(\frac{1}{\gamma}\right) + \gamma - \frac{\pi}{2}. \tag{14}$$

The function $W(\gamma)$ is shown in Fig. 1 and has the asymptotic properties

$$W(\gamma) \simeq \begin{cases} -\pi\, \theta(-\gamma) + \pi \operatorname{sgn}(\gamma) \frac{\gamma^2}{2} , & \gamma \to 0 , \\ \\ 2\,\gamma & \gamma \to \pm\infty . \end{cases} \tag{15}$$

In Section 5 we investigate what happens when the impurity is placed at $x_1$ to the right of the edge $x_e$, such that $V(x_1) > V(x_e) = \mu$. When the impurity is attractive we show that a phase transition occurs as the reduced strength $\lambda_A = -\lambda > 0$ of the impurity is increased beyond a critical value. We call this transition *a filling transition*. Let us recall from elementary quantum mechanics of a single particle that a delta potential introduced at $x_1$, in addition to a flat potential $V(x) = V_0$, introduces a single bound

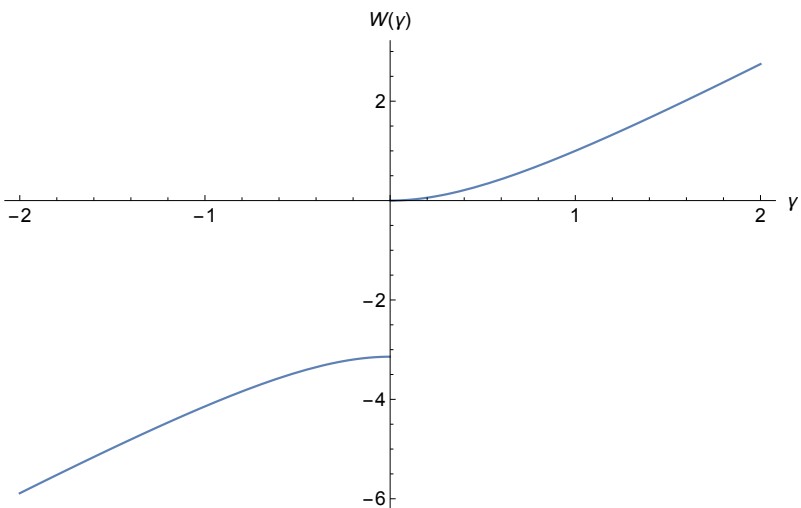

Figure 1: The scaling function for the effective potential $W(\gamma)$ defined in Eq. (14). Note that although $W$ is discontinuous at $\gamma = 0$ the potential $V_{\text{eff}}$ is continuous. Note that although $W$ is discontinuous at $\gamma = 0$, the potential $V_{\text{eff}}(x_1)$ is continuous as a function of $\lambda$ at $\lambda = 0$.

state with wave function $\psi_b(x) = \sqrt{\lambda_A} \exp(-\lambda_A |x - x_1|)$ and energy $E_b = -\frac{\hbar^2 \lambda_A^2}{2m} + V_0$. Substituting $V_0 = V(x_1)$ we find that there are two different phases: (i) weak impurity, where $E_b > \mu$ implying that this bound state is unoccupied and consequently $\rho_\mu(x_1) \simeq 0$; (ii) strong impurity, where $E_b < \mu$ implying that this bound state is occupied and consequently $\rho_\mu(x_1) \simeq \lambda_A$. The transition occurs exactly at $E_b = \mu$, which corresponds to $\lambda_A = \kappa_\mu(x_1) = \sqrt{2(V(x_1) - \mu)}$. This filling transition has some similarities with the BBP transition in random matrix theory, where a rank one perturbation to a random matrix can displace the maximal eigenvalue of the matrix [17, 18].

In Section 6 we study the effect of adding two impurities in the bulk, say at $x_1$ and $x_2$, close to $x = 0$. We assume that $x_1$ and $x_2$ are such that $|V(x_1) - V(x_2)| \ll E_F = \hbar^2 k_F^2 / (2m)$ where $k_F = k_F(0)$ is the Fermi wave vector at $x = 0$. This condition ensures that the potential remains effectively constant on the scale of the separation $r = |x_1 - x_2|$ between the impurities. We show that the effective Casimir-like interaction between these two impurities, mediated by the background Fermi gas, is given by

$$
\begin{aligned}
V_{\text{int}}(r, k_F, \gamma_1, \gamma_2) &= -\frac{2E_F}{\pi \zeta} \text{Re} \int_0^\infty ds \left( 1 - i\frac{s}{\zeta} \right) \\
&\times \ln \left( 1 + \frac{\gamma_1 \gamma_2}{[1 - i\frac{s}{\zeta} - i\gamma_1][1 - i\frac{s}{\zeta} - i\gamma_2]} \exp(-2\zeta i - 2s)) \right) , \quad (16)
\end{aligned}
$$

where $\zeta = k_F r$ and $\gamma_i = \lambda_i / k_F$ are the scaled impurity strengths. Our result is valid for all $\zeta$. Interestingly, using free fermionic field theory, an expression for $V_{\text{int}}(r, k_F, \gamma_1, \gamma_2)$ was derived in Refs. [13, 14], which reads

$$
V_{\text{int}}(r, k_F, \gamma_1, \gamma_2) \simeq \frac{E_F}{\pi \zeta} \text{Re} \, \text{Li}_2 \left( -\frac{\gamma_1 \gamma_2}{[1 - i\gamma_1][1 - i\gamma_2]} \exp(-2i\zeta) \right) , \quad (17)
$$

where $\text{Li}_2(z) = \sum_{n=1}^\infty z^n / n^2$ is the di-logarithm function and Re denotes the real part. Our formula (16) can be shown to reduce to (17) when $\zeta \gg 1$. However, this form (17) is an approximate form that holds only for large $\zeta$. As $\zeta \to 0$, $V_{\text{int}}(r, k_F, \gamma_1, \gamma_2)$ in Eq. (17)

diverges, which is not physical. Instead, our exact result (16), which holds for all $\zeta$, approaches to a constant as $\zeta \to 0$ (see Fig. 6).

In Section 7 we analyse the effect due to an impurity close to the edge of the Fermi gas, i.e. $x_1 \approx x_e$. The Friedel oscillations [9–11] around impurities in the bulk are strongly suppressed near the edge. However weak oscillations, that are already present at the edge without any impurities, still persist in the presence of impurities. The main effect of the impurity is to alter the phase of these oscillations. For an attractive impurity, we show that the filling transition discussed above becomes a smooth crossover on the scale of the inter-particle distance at the edge and the local density profile is described by a universal scaling function. Finally we obtain an analytic expression for the effective potential acting on the impurity placed in the edge region.

In Section 8 present our general conclusions and perspectives for future studies.

## 3 Basic formalism and set up

We now describe how the single particle Green's function can be used to extract the change in the kernel due to the addition of an impurity at a fixed position in the system. The results given below are derived in detail in a recent paper [8] and we refer the reader there for detailed derivations.

### 3.1 Kernels via Green's functions

The Green's function $G_{\mu'}(x, y)$ associated to the Hamiltonian $H$ in (1) is defined for an arbitrary running Fermi energy $\mu'$ as

$$G_{\mu'}(x, y) = \sum_k \frac{\psi_k^*(x)\psi_k(y)}{\mu' - i0^+ - \epsilon_k} \ . \tag{18}$$

The Green's function has poles at $\mu' = \epsilon_k + i0^+$, i.e., infinitesimally above the real axis in the complex plane. In operator notation we also have the equivalent resolvent representation

$$G_{\mu'} = (\mu' - i0^+ - H)^{-1}, \tag{19}$$

from which we see that $G_{\mu'}(x, y)$ is solution to the equation

$$\frac{\hbar^2}{2m}\frac{\partial^2}{\partial x^2}G_{\mu'}(x, y) + (\mu' - i0^+ - V(x))G_{\mu'}(x, y) = \delta(x - y). \tag{20}$$

The kernel can be obtained from the Green's function from the following formula

$$K_\mu(x, y) = \frac{1}{\pi}\int_{-\infty}^{\mu} d\mu' \text{Im}\, G_{\mu'}(x, y) = \frac{1}{\pi}\text{Im}\int_{-\infty}^{\mu} d\mu'\, G_{\mu'}(x, y), \tag{21}$$

where the imaginary part can be taken outside the integral as the integration contour is real. Noting that when $\mu \to \infty$ the kernel becomes the sum over a complete set of states, we can derive an alternative representation

$$K_\mu(x, y) = \delta(x - y) - \frac{1}{\pi}\text{Im}\int_{\mu}^{\infty} d\mu'\, G_{\mu'}(x, y). \tag{22}$$

In this paper we will be interested in the change in the kernel due to an added impurity. If we denote the Green's function in the absence of impurities by $G_{0\mu}(x, y)$, then we can write the Green's function as $G_\mu(x, y) = G_{0\mu}(x, y) + \Delta G_\mu(x, y)$ where $\Delta G_\mu(x, y)$ is the

change in the Green's function due to the impurities. It is then easy to see that the change in kernel $\Delta K_\mu(x, y) = K_\mu(x, y) - K_{0\mu}(x, y)$ is given by

$$\Delta K_\mu(x, y) = \frac{1}{\pi} \mathrm{Im} \int_{-\infty}^{\mu} d\mu' \, \Delta G_{\mu'}(x, y), \tag{23}$$

if one uses Eq. (21), or alternatively

$$\Delta K_\mu(x, y) = -\frac{1}{\pi} \mathrm{Im} \int_{\mu}^{\infty} d\mu' \, \Delta G_{\mu'}(x, y) \tag{24}$$

if one uses Eq. (22). The fact that these two representations (23) and (24) are equivalent can also be seen from the following argument. These integrals can be interpreted as contour integrals in the complex $\mu'$ plane along the real axis. In Fig. 2 these two contours are represented as $\Gamma_1 = (\mu, \infty)$ for (24) and $\Gamma_2 = (-\infty, \mu)$ for (23), along with the position of the poles which are infinitesimally above the real axis and are shown by crosses. In terms of the contours $\Gamma_2$ and $\Gamma_1$ shown on the figure we have $\Delta K_\mu(x, y) = \frac{1}{\pi} \mathrm{Im} \int_{\Gamma_1} d\mu' \, \Delta G_{\mu'}(x, y) = -\frac{1}{\pi} \mathrm{Im} \int_{\Gamma_2} d\mu' \, \Delta G_{\mu'}(x, y)$. The equivalence of the two representations can also be demonstrated as follows. First, using Cauchy's theorem which, as there are no poles in the lower half of the complex plane, gives $\int_{\Gamma_1 \cup \Gamma_2 \cup \Gamma_3} d\mu' \, \Delta G_{\mu'}(x, y) = 0$, where $\Gamma_3$ is taken to be an infinite semicircle, with center at the origin, in the lower half of the complex plane. One can then show that $\int_{\Gamma_3} d\mu' \, \Delta G_{\mu'}(x, y) = 0$ to obtain the desired result. Finally we should point out that by rotating the contour $\Gamma_1$ about $z = \mu$ by $-\pi/2$ (so it is parallel to the imaginary axis) gives an integral representation that corresponds to the sum over Matsubara frequencies in the fermionic field theory setting [13, 14].

It is important to note here that the representation given in Eq. (24) has a number of advantages over that in Eq. (23). First, it involves an integral over $\mu' > \mu$ therefore we do not need to know the Green's function $\Delta G_{\mu'}(x, y)$ for small $\mu' < \mu$. Since $\mu$ is large, we just need to know the Green's function for large $\mu'$ which can be conveniently computed using the semi-classical approximation. Furthermore, we will see that the representation in Eq. (24) is more suitable to asymptotic analysis of certain formulas.

## 3.2 Bulk and edge Green's function

In this section, we recall the results obtained in Ref. [8] for the kernel both in the bulk as well as at the edges, using the Green's function method, for a smoothly varying trapping potential $V(x)$.

**In the bulk**. We start with the bulk and consider the potential around a point $x_0$ where we assume that $V(x) \approx V(x_0)$ and solve Eq. (20) with a constant potential $V(x_0)$. This gives

$$G_{\mu'}(x_0 + z, x_0 + z') = \frac{im}{\hbar^2} \frac{\exp(-ik_{\mu'}(x_0)|z - z'|)}{k_{\mu'}(x_0)}, \tag{25}$$

where

$$k_{\mu'}(x_0) = \sqrt{2m(\mu' - V(x_0))}/\hbar - i0^+ \tag{26}$$

is the local Fermi wave vector given the Fermi energy $\mu'$. The positive value of the square root is taken and it is understood to have a negative infinitesimal imaginary part as indicated above. By inserting this expression (25) in Eq. (21), setting $x = y$ and performing the integral over $\mu'$ one finds the density in the bulk

$$\rho_\mu(x) = \frac{\sqrt{2m(\mu - V(x))}}{\pi\hbar} = \frac{k_\mu(x)}{\pi} \ . \tag{27}$$

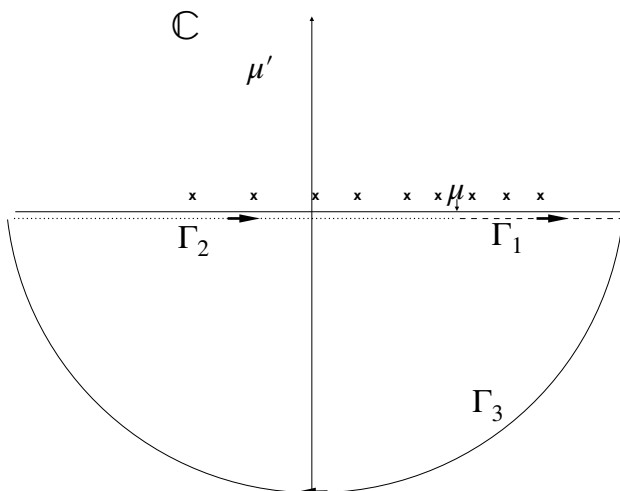

Figure 2: Contour integrals used in the integral representations of the kernel. Crosses, ×, indicate the poles of the Green's function which are just above the real axis. $\Gamma_1 = (\mu, \infty)$ is the contour used in representation Eq. (23) and $\Gamma_2 = (-\infty, \mu)$ that is used for representation Eq. (24). The contour $\Gamma_3$ is used to close the contour $\Gamma_1 \cup \Gamma_2$ and is taken to be a semi-circle in the lower half of the complex plane whose radius is taken to $\infty$.

These results are obviously exact for a flat potential $V(x) = V_0$. However they are also accurate as long as the relative variations of $k_\mu(x)$ on microscopic scales, of order $O(1/k_\mu)$ are negligible, i.e.,

$$\left| k_\mu[x_0 + 1/k_\mu(x_0)] - k_\mu(x_0) \right| \approx \left| \frac{k'_\mu(x_0)}{k_\mu(x_0)} \right| \ll k_\mu(x_0) . \tag{28}$$

Using $k'_\mu(x_0) \propto V'(x_0)/k_\mu(x_0)$ from Eq. (27), this condition (28) translates to [8]

$$R = \frac{\hbar |V'(x_0)|}{m^{\frac{1}{2}} |2\mu - 2V(x_0)|^{\frac{3}{2}}} \ll 1 . \tag{29}$$

Note that this argument naturally introduces a length scale

$$\xi = \frac{k_\mu(x_0)}{k'_\mu(x_0)} \tag{30}$$

which sets the size of the region over which this assumption that $V(x)$ is constant holds. The condition in (29) clearly gets violated in two cases: (i) when the potential is not smooth, for instance when the potential exhibits a step structure as studied in [8] or when there is a delta function contribution to the potential (ii) when the analysis is carried out at the *edge* of the trap where the density in Eq. (27) vanishes.

**At the edge.** The edge $x_e$ of the Fermi gas occurs where the density vanishes, i.e. when

$$\mu = V(x_e) . \tag{31}$$

We see from Eq. (29) that $R$ diverges as $x_0 \to x_e$. The physics near the edge region (the so called *Airy gas*), can be studied by linearizing the potential $V(x)$ near $x = x_e$, i.e.

$$V(x_e + z) \simeq V(x_e) + zV'(x_e) . \tag{32}$$

In this region, by solving Eq. (20) with a linear potential (32), the Green's function can be written in the scaling form [8]

$$G_{\mu'}(x_e + z, x_e + z') = \frac{1}{\alpha_e w_e} g_e\left(\frac{z}{w_e} + \frac{V(x_e) - \mu'}{\alpha_e}, \frac{z'}{w_e} + \frac{V(x_e) - \mu'}{\alpha_e}\right), \tag{33}$$

where $w_e$ and $\alpha_e$ are respectively the length scale associated with $1/\rho_e$, where $\rho_e$ is the fermion density at the edge [5] and the energy scale associated with the edge given by

$$w_e = \left(\frac{\hbar^2}{2mV'(x_e)}\right)^{\frac{1}{3}} , \quad \alpha_e = \left(\frac{\hbar^2 V'(x_e)^2}{2m}\right)^{\frac{1}{3}} = V'(x_e) w_e . \tag{34}$$

The function $g_e(\zeta, \zeta')$ is given by

$$\begin{aligned} g_e(\zeta, \zeta') &= -\pi \mathrm{Ai}(\zeta)[-i\mathrm{Ai}(\zeta') + \mathrm{Bi}(\zeta')] \text{ for } \zeta > \zeta' \tag{35} \\ &= -\pi \mathrm{Ai}(\zeta')[-i\mathrm{Ai}(\zeta) + \mathrm{Bi}(\zeta)] \text{ for } \zeta < \zeta' . \tag{36} \end{aligned}$$

For later purposes, we note that

$$\mathrm{Im}\left[g_e(\zeta, \zeta')\right] = \pi \mathrm{Ai}(\zeta)\mathrm{Ai}(\zeta'). \tag{37}$$

**Outside the bulk**. In the classically forbidden region, far outside the edge where the condition in (29) holds, we will again assume that $V(x)$ is slowly varying around a point $x_0$. In this case, the solution of Eq. (20) reads

$$G_{\mu'}(x_0 + z, x_0 + z') \approx -\frac{m}{\hbar^2} \frac{\exp(-(\kappa_{\mu'}(x_0) + i0^+)|z - z'|)}{[\kappa_{\mu'}(x_0) + i0^+]}, \tag{38}$$

where

$$\kappa_{\mu'}(x_0) = \frac{\sqrt{2m(V(x_0) - \mu')|}}{\hbar} \tag{39}$$

is positive. This result is similar to the one found in the bulk in Eq. (25) with a local Fermi wavevector which is imaginary.

One can check, using the asymptotic properties of the Airy functions $\mathrm{Ai}(z)$ and $\mathrm{Bi}(z)$, that the edge result for the Green's function (33) matches (i) on the left with the bulk result (20) and (ii) on the right with the result far outside the bulk (38).

### 3.3 Treating delta function potentials

In this section we show how one can use the Green's function method to study systems where the potential has a delta-function part. The delta function potential has been extensively studied in quantum systems to model impurities [29]. Here we show that the Green's function method is particularly well suited to study this problem.

We start with a single particle Hamiltonian $H_0$ and denote its Green's function by

$$G_{0\mu} = (\mu - i0^+ - H_0)^{-1} . \tag{40}$$

We now add $n$ impurities so that the total Hamiltonian is

$$H = H_0 + \Delta H, \tag{41}$$

with

$$\Delta H = \sum_{i=1}^{n} g_i \delta(x - x_i). \tag{42}$$

The coordinates $x_i$'s are the positions of the impurities and $g_i$'s denote their interaction strengths with the fermions. Note that the sign of $g_i$ can be positive (repulsive) or negative (attractive). Many methods have been found [29] to extract the Green's function $G_\mu = (\mu - i0^+ - H)^{-1}$. A simple way to do this is to observe that

$$G_\mu(x,y) = G_{0\mu}(x,y) + \sum_{i=1}^{n} A_i G_{0\mu}(x,x_i) , \qquad (43)$$

yields a solution to $[\mu - i0^+ - H]G_\mu = \delta(x-y)$ if the $A_i$'s obey the linear equations.

$$A_i - g_i G_{0\mu}(x_i,y) - \sum_{j=1}^{n} g_i A_j G_{0\mu}(x_i,x_j) = 0 . \qquad (44)$$

Note that the $A_i$'s depend implicitly on both the $x_i$'s and $y$. The solution of this linear equation (44) can be expressed as

$$A_i = \sum_{j=1}^{n} R_{ij}^{-1} g_j G_{0\mu}(x_j,y) \qquad (45)$$

where the $n \times n$ matrix $R$ has components $R_{ij}$ given by

$$R_{ij} = \delta_{ij} - g_i G_{0\mu}(x_i,x_j). \qquad (46)$$

This leads to the change in the Green's function

$$\Delta G_\mu(x,y) = \sum_{i,j=1}^{n} R_{ij}^{-1} g_j G_{0\mu}(x,x_i) G_{0\mu}(x_j,y) . \qquad (47)$$

For general $x$ and $y$ the above expression is quite complicated. However, if we consider the Green's function at points where there are impurities and define the $n \times n$ matrices $\mathcal{G}_0$ and $\mathcal{G}$ such that $\mathcal{G}_{0ij} = G_{0\mu}(x_i,x_j)$ and $\mathcal{G}_{ij} = G_\mu(x_i,x_j)$ things simplify a bit. In this case, using Eq. (47) and adding to $G_{0\mu}$, one gets in the matrix form

$$\mathcal{G} = \mathcal{G}_0(I + R^{-1}\Lambda_g \mathcal{G}_0) \qquad (48)$$

where the matrix $\Lambda_g$ has components $\Lambda_{gij} = g_i \delta_{ij}$. Noting that Eq. (46) implies $R + \Lambda_g \mathcal{G}_0 = I$ and multiplying both sides by $R^{-1}$ gives $I + R^{-1}\Lambda_g \mathcal{G}_0 = R^{-1}$. Using this result, Eq. (48) now reads

$$\mathcal{G} = \mathcal{G}_0 R^{-1} = \mathcal{G}_0(I - \Lambda_g \mathcal{G}_0)^{-1} , \qquad (49)$$

where we further used Eq. (46) for $R$. Thanks to this relation, one can compute $\mathcal{G}$ if one knows $\mathcal{G}_0$.

Using, furthermore, the formula for the derivative of the determinant of a matrix with respect to a parameter $t$, *i.e.*

$$\frac{\partial}{\partial t} \ln[\det A(t)] = \text{Tr}\left[A^{-1}(t)\frac{\partial}{\partial t}A(t)\right], \qquad (50)$$

we can rewrite Eq. (49) as

$$G_\mu(x_i,x_i) = -\frac{\partial}{\partial g_i} \ln\left(\det[1 - \Lambda_g \mathcal{G}_0]\right) . \qquad (51)$$

We will use this representation later.

Let us consider the simplest case of a single impurity located at $x_1$ with an amplitude $g_1 = g$ (this corresponds to $n = 1$ in Eq. (42)). In this case, the full Green's function can be computed from Eq. (43), (45) and (46)

$$G_\mu(x,y) = G_{0\mu}(x,y) + \frac{g\, G_{0\mu}(x,x_1) G_{0\mu}(x_1,y)}{1 - g G_{0\mu}(x_1,x_1)} \ , \tag{52}$$

which has a simple "Schwinger-Dyson" form. Expanding the denominator in powers of $g$, this formula (52) has a simple physical interpretation: it adds up contributions to the Green's function arising from no scattering, one scattering, two scatterings, etc, from the impurity. Exactly at the position of the impurity it reads

$$G_\mu(x_1,x_1) = \frac{G_{0\mu}(x_1,x_1)}{1 - g G_{0\mu}(x_1,x_1)} \ . \tag{53}$$

We will see below that this formula is particularly useful to deduce the fermion density at the impurity as well as the energy change induced by the introduction of an impurity.

In the following, we will use the formulae derived in this section in the case where $H_0$ corresponds to a smooth trapping potential in the absence of a delta-potential.

## 4 Impurity in the bulk

Here we consider the effects of delta-function impurities in the bulk. The overall smooth trapping potential $V(x)$ described by $H_0$ (see Eq. (1)), as discussed before, can be taken to be locally constant and without loss of generality we set $V(x) = 0$. Thus the Fermi wave vector at Fermi energy $\mu$ is given by $k_F = k_\mu = \sqrt{2m\mu}/\hbar$. The local density of the system without impurity is given by $\rho_{0\mu} = \frac{k_F}{\pi}$.

The effect of impurities has been well studied in the literature, notably the density around impurities exhibits the well known Friedel oscillations [9–11]. However, in most previous studies only the behavior of the density at distances greater than the interparticle distance $\ell_0 = 1/\rho_{0\mu}$ or in the linear response regime (i.e. to first order in $\lambda$). In [12] the density change induced by a delta function impurity was studied exactly. Here, by using a more versatile method, we show how this result can be rigorously extended to inhomogeneous bulk systems. We also show how the result given in [12] is, simply, modified to take properly into account the appearance of a bound state in the case of an attractive impurity.

It is important to know the exact behavior of this density at the location of the impurity since, as we will see, it determines the effective interaction between the impurity and the surrounding fermions. We show that this density at the impurity is finite and depends on the local Fermi-wave vector. We also compute the effective interaction and show that it is given by Eq. (14).

### 4.1 Friedel oscillations

We start by computing the kernel in the presence of a single impurity of strength $g_1 = g$ placed at $x_1 = 0$. Here the change in the Green's function, using Eqs. (24), (25) and (52) for the bulk Green's function yields a change in the kernel $\Delta K_\mu(x,y) = K_\mu(x,y) - K_{0\mu}(x,y)$ which is given by

$$\Delta K_\mu(x,y) = \frac{g m^2}{\pi \hbar^2} \text{Im} \int_\mu^\infty d\mu' \, \frac{\exp(-i k_{\mu'}[|x| + |y|])}{\hbar k_{\mu'}(\hbar k_{\mu'} - i\frac{gm}{\hbar})}. \tag{54}$$

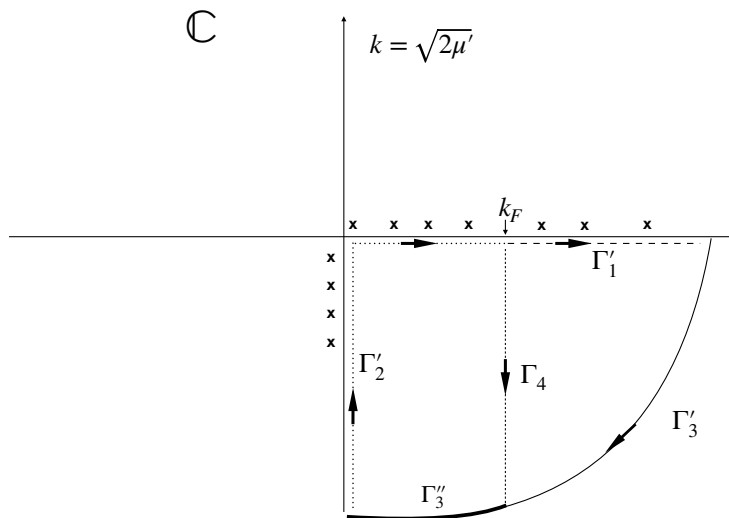

Figure 3: Contour integrals used in the integral representations of the kernel in terms of the variable $k = \sqrt{2m\mu'/\hbar^2}$. Crosses indicate the poles of the Green's function in the complex plane of $k$. The contours $\Gamma_i'$ for $i = 1, 2, 3$ correspond to the contours $\Gamma_i$ shown in Fig. 2 when mapped into the $k$ plane. Note that the contour $\Gamma_2'$ has two components: one vertical, and one horizontal going from $k = 0$ to $k = k_F$. The contour $\Gamma_3''$ denotes a portion on $\Gamma_3'$, shown in bold, which is useful for later purpose. The contour $\Gamma_4 = (k_F, k_F - i\infty)$, which is useful for asymptotic analysis, is also shown.

Now we change variables $\mu' = \hbar^2 k/2m$ (so $d\mu' = \hbar^2 k dk/m$) and we assume that $\mu > 0$ so that the integration over $k$ is along the real axis. The way in which all the contours in Fig. 2 transform under the transformation $k = \sqrt{\frac{2m}{\hbar^2}\mu'}$ is shown in Fig. 3 along with the position of the original poles in the Green's function shown again as crosses. This gives

$$\Delta K_\mu(x,y) = \frac{\lambda}{\pi} \text{Im} \int_{k_F}^\infty dk \frac{\exp(-ik[|x| + |y|])}{k - i\lambda}, \tag{55}$$

where

$$\lambda = mg/\hbar^2 \tag{56}$$

is an inverse length scale associated with the impurity, and $k_F = k_F(0) = \sqrt{2m\mu}/\hbar$ is the Fermi wave vector at the position of the impurity.

The integral in Eq. (55) corresponds to the contour $\Gamma_1'$. The contour $\Gamma_1'$ can be deformed onto the contour $\Gamma_4$ as the integral over the contour $\Gamma_3'$ is zero and there are no poles crossed during this deformation [8], to give

$$\Delta K_\mu(x,y) = \frac{\lambda}{\pi} \text{Im} \int_0^\infty -id\kappa \frac{\exp(-(ik_F + \kappa)[|x| + |y|])}{k_F - i\kappa - i\lambda}. \tag{57}$$

The above can also be written in terms of standard functions via the change of variables $s' = \kappa + \lambda + ik_F$ to obtain

$$\begin{aligned}
\Delta K_\mu(x,y) &= \frac{\lambda \exp(\lambda(|x| + |y|))}{\pi} \text{Im} \int_{\lambda + ik_F}^\infty \frac{ds'}{s'} \exp(-(|x| + |y|)s') \\
&= \frac{\lambda \exp(\lambda(|x| + |y|))}{\pi} \text{Im } E_1[(\lambda + ik_F)(|x| + |y|)]
\end{aligned} \tag{58}$$

where we recall that

$$\mathrm{E}_1(\zeta) = \int_\zeta^\infty dt \, \frac{\exp(-t)}{t} \tag{59}$$

is the exponential integral function [28].

Taking $x = y$ we find that the change in the local density $\Delta\rho_\mu(x)$ around the position of the impurity at $x = 0$ is given by

$$\Delta\rho_\mu(x) = \frac{\lambda \exp(2\lambda|x|)}{\pi} \mathrm{Im} \, \mathrm{E}_1(2(\lambda + ik_F)|x|) \, . \tag{60}$$

This is an important result of this paper, since it is valid for all values of $\lambda$ and at all distances $x$. The comparison between this result (60) and the result of Ref. [12] is performed in Appendix A. We now study this form of the density profile in Eq. (60) at distances respectively very close and very far from the impurity.

**Density close the impurity**. Here we analyse the density in Eq. (60) very close to the impurity, i.e. the limit $x \to 0$. For small $x$ we can use the asymptotic expansion $\mathrm{E}_1(x) = -\gamma_E - \ln(x) + x + O(x^2)$ at small $x$ [28]. In particular at $x = 0$, taking the imaginary part of the logarithm, we get the total local density (with the bulk term $\rho_{0\mu} = k_F/\pi$ included)

$$\rho_\mu(0) = \frac{k_F}{\pi} - \frac{\lambda}{\pi} \arg(ik_F + \lambda) = \frac{k_F}{\pi} - \frac{\lambda}{2} + \frac{\lambda}{\pi} \tan^{-1}\left(\frac{\lambda}{k_F}\right), \tag{61}$$

where $\tan^{-1}$ above is the principle branch such that $\tan^{-1}(0) = 0$. The change in the density is equal to zero when $\lambda = 0$ as because $k_F \equiv k_F - i0^+$ one has that $\arg(ik_F) = \pi/2$. The change in the local density is significant if $\lambda \sim k_F$.

**Repulsive case.** In the limit of strong repulsion if $\lambda = \lambda_R$ with $\lambda_R > 0$ and $\lambda_R \gg k_F$ we find

$$\rho_\mu(0) \simeq \frac{k_F^3}{3\lambda_R^2 \pi} \, , \tag{62}$$

while for $\lambda_R \ll k_F$ one has

$$\rho_\mu(0) \simeq \frac{k_F}{\pi} - \frac{\lambda_R}{2}. \tag{63}$$

**Attractive case.** In this case, setting $\lambda = -\lambda_A$ with $\lambda_A > 0$, we find, for strong attraction $\lambda_A \gg k_F$

$$\rho_\mu(0) \simeq \lambda_A + \frac{k_F^3}{3\lambda_A^2 \pi} \, . \tag{64}$$

Note that the dominant term $\lambda_A$ in Eq. (64) is the contribution from the single bound state associated to the attractive delta function. Indeed such a bound state has a wave function

$$\psi_b(x) = \sqrt{\lambda_A} \exp(-\lambda_A|x|) \tag{65}$$

which gives rise to a density $|\psi_{bs}(0)|^2 = \lambda_A$. On the other hand, for weak attraction $\lambda_A \ll k_F$

$$\rho_\mu(0) \simeq \frac{k_F}{\pi} + \frac{\lambda_A}{2}. \tag{66}$$

**Density far from the impurity and Friedel Oscillations.** In this case the asymptotic expansion [28] for $|\zeta| \gg 1$ and $|\arg(\zeta)| < 3\pi/2$,

$$\mathrm{E}_1(\zeta) \sim \zeta^{-1} \exp(-\zeta), \tag{67}$$

can be used but one can also use the representation given in Eq. (57) where the long distance behavior comes from an expansion about $\kappa = 0$. We find from (60) that the density for large $x$ decays as

$$\Delta\rho_\mu(x) \simeq -\frac{\lambda}{2\pi|x|(\lambda^2 + k_F^2)}\left[k_F\cos(2k_F|x|) + \lambda\sin(2k_F|x|)\right], \tag{68}$$

which can be rewritten as

$$\Delta\rho_\mu(x) \simeq -\frac{\lambda}{2\pi|x|\sqrt{\lambda^2 + k_F^2}}\sin\left(2k_F|x| + \tan^{-1}\left(\frac{k_F}{\lambda}\right)\right). \tag{69}$$

At large $|x|$, both the repulsive and attractive cases are described by the formula (69) with $\lambda = \lambda_R > 0$ in the repulsive case while $\lambda = -\lambda_A < 0$ in the attractive case. In the limit of small $\lambda$, one finds

$$\Delta\rho_\mu(x) \simeq -\frac{\lambda}{2\pi k_F|x|)}\cos(2k_F|x|), \tag{70}$$

which is the standard formula for large distance one-dimensional Friedel oscillations in the regime of linear response [12].

In Fig. 4 we plot the relative perturbation of the exact density

$$\frac{\Delta\rho_\mu(x)}{\rho_{0\mu}} = n(\zeta, \gamma) = \gamma\exp(2\gamma|\zeta|)\mathrm{Im}\,\mathrm{E}_1(2(\gamma + i)|\zeta|)) \tag{71}$$

as a function of $\zeta = k_F x$ where we recall that $\rho_{0\mu}$ is the local density in the absence of the impurity and where we have written $\lambda = \gamma k_F$ [30]. In Fig. 4 we plot $n(\zeta, \gamma)$ in the repulsive case with $\gamma = 1$. The asymptotic expansion Eq. (69) is also shown as an orange dashed line. For $\zeta > 3$, this asymptotic form describes accurately the exact result. However, when extrapolated to small values of $\zeta$, this asymptotic form diverges while the exact result approaches a finite value as $\zeta \to 0$ (see Eq. (61)). In Fig. 5 we plot $n(\zeta, \gamma)$ for an attractive impurity with $\gamma = -1$. We see the signature of the localized wave function about the impurity which causes an increase in the local density. Again we see that the asymptotic approximation for Friedel oscillations becomes accurate only for $\zeta \sim 3$.

Going beyond the density we can also analyse the kernel $K_\mu(x, y)$ in Eq. (58) for large $|x|$ and $|y|$ by using the same asymptotics for the Exponential integral. We find at large $|x|$ and $|y|$, both for the repulsive and attractive cases,

$$\Delta K_\mu(x, y) \simeq -\frac{\lambda}{\pi(|x| + |y|)\sqrt{\lambda^2 + k_F^2}}\sin\left(k_F(|x| + |y|) + \tan^{-1}\left(\frac{k_F}{\lambda}\right)\right). \tag{72}$$

## 4.2 The effective potential acting on an impurity

Here we examine how the total energy of the fermion system is changed by adding of a single impurity at fixed Fermi energy $\mu$. The Hamiltonian $H(\lambda)$ depends explicitly on the parameter $\lambda = gm/\hbar^2$ where $g$ is the impurity strength. We denote by $\epsilon_k(\lambda)$ the $k^{\text{th}}$ energy level, as a function of $\lambda$. We define the total energy $E(\mu, \lambda)$ and the total number of fermions $N(\mu, \lambda)$ as

$$E(\mu, \lambda) = \sum_k \theta(\mu - \epsilon_k(\lambda))\epsilon_k(\lambda) \quad, \quad N(\mu, \lambda) = \sum_k \theta(\mu - \epsilon_k(\lambda)). \tag{73}$$

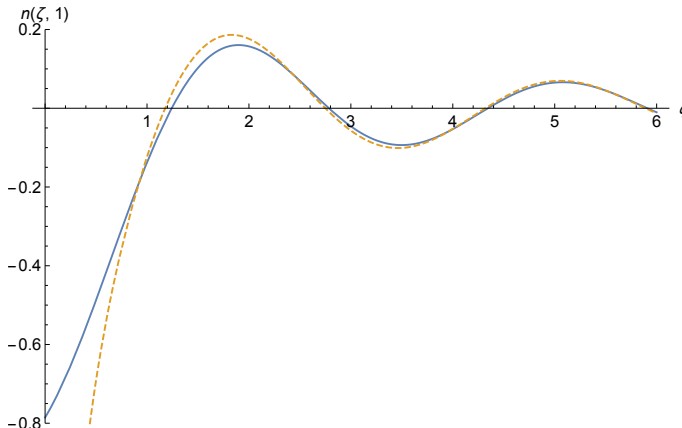

Figure 4: The relative change in the density around the point $x = x_1$, given by Eq. (71) due to a repulsive delta function potential at the point $x = 0$ of amplitude $\lambda_R = \gamma k_F$ for $\gamma = 1$ (solid line), and its asymptotic approximation (dashed line) obtained using Eq. (69)

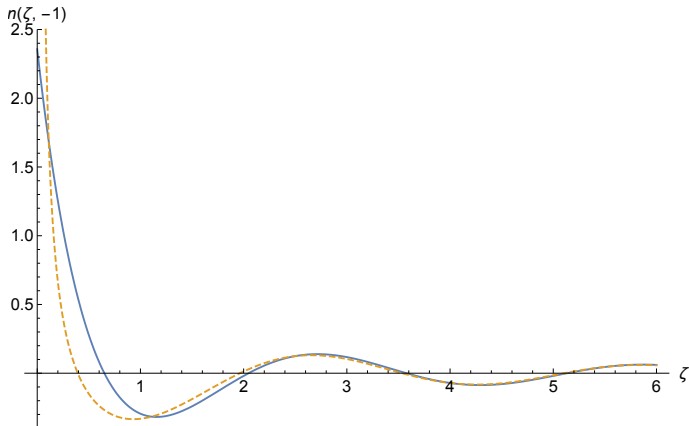

Figure 5: The relative change in the density around the point $x = x_1$, given by Eq. (71) due to an attractive delta function potential at the point $x = 0$ of amplitude such that $\lambda = -\lambda_A = \gamma k_F$ for $\gamma = -1$ (solid line), and its asymptotic approximation (dashed line) using Eq. (69).

The goal is to compute the effective potential felt by the impurity at position $x_1$ which can be identified as the change in the grand-potential (since we are working at fixed Fermi energy $\mu$)

$$V_{\text{eff}}(x_1) = \Omega(\mu, \lambda) - \Omega(\mu, 0) \qquad \text{where} \qquad \Omega(\mu, \lambda) = E(\mu, \lambda) - \mu N(\mu, \lambda) . \tag{74}$$

This problem for a homogeneous system, where the potential is constant, has been studied in [22, 23].

To perform this computation, we use the Hellmann-Feynman theorem which states that

$$\frac{\partial \epsilon_k(\lambda)}{\partial \lambda} = \int dx \ \psi_k^*(x, \lambda) \frac{\partial H}{\partial \lambda} \psi_k(x, \lambda) , \tag{75}$$

where $\psi_k(x, \lambda)$ is the eigenstate associated to the energy level $\epsilon_k(\lambda)$. In the present case

of the delta-impurity this theorem (75) gives

$$\frac{\partial \epsilon_k(\lambda)}{\partial \lambda} = \frac{\hbar^2}{m} |\psi_k(x_1, \lambda)|^2. \tag{76}$$

From this, one sees that every energy level is moved up for repulsive impurities and down for attractive ones. Thus at fixed $\mu$, the derivatives of the energy $E(\mu, \lambda)$ and of the number of particles $N(\mu, \lambda)$ in Eq. (73) with respect to $\lambda$ read

$$\partial_\lambda E(\mu, \lambda) = \sum_k \theta(\mu - \epsilon_k) \partial_\lambda \epsilon_k(\lambda) - \mu \sum_k \partial_\lambda \epsilon_k(\lambda) \delta(\mu - \epsilon_k(\lambda)) \tag{77}$$

$$\partial_\lambda N(\mu, \lambda) = -\sum_k \partial_\lambda \epsilon_k(\lambda) \delta(\mu - \epsilon_k(\lambda)) . \tag{78}$$

Therefore, using Eq. (74), the derivative of $\Omega(\mu, \lambda)$ with respect to $\lambda$, using (77) and (78) together with (75) is given by

$$\partial_\lambda \Omega(\mu, \lambda) = \frac{\hbar^2}{m} \sum_k \theta(\mu - \epsilon_k(\lambda)) |\psi_k(x_1, \lambda)|^2 = \frac{\hbar^2}{m} \rho_\mu(x_1, \lambda) , \tag{79}$$

where we have made explicit the dependence of the fermion density on $\lambda$. The effective interaction of the impurity with the fermion system is thus given by

$$V_{\text{eff}}(x_1) = \Omega(\mu, \lambda) - \Omega(\mu, 0) = \frac{\hbar^2}{m} \int_0^\lambda d\lambda' \rho_\mu(x_1, \lambda') . \tag{80}$$

We can now use the expression for the fermion density given in Eq. (61) to obtain

$$V_{\text{eff}}(x_1) = \frac{\hbar^2}{2\pi m} \left[ (k_F^2(x_1) + \lambda^2) \tan^{-1}\left(\frac{\lambda}{k_F(x_1)}\right) + k_F(x_1)\lambda - \frac{\pi\lambda^2}{2} \right] = \frac{\hbar^2 \lambda^2}{2\pi m} W\left(\frac{k_F(x_1)}{\lambda}\right), \tag{81}$$

where the function $W(\gamma)$ is given in Eq. (14). When $k_F(x)$ is constant, the formula Eq. (81) agrees with that found in [23] for homogeneous systems.

In the limit of a weak impurity strength $|\lambda| \ll k_F(x_1)$ we find from (81)

$$V_{\text{eff}}(x_1) \simeq \frac{\hbar^2 \lambda \, k_F(x_1)}{\pi m} . \tag{82}$$

Therefore for $\lambda > 0$ the impurity is pushed away from a dense region, while for $\lambda < 0$ it is attracted by dense regions. On the other hand, for a strong impurity strength, $|\lambda| \gg k_F(x_1)$, we find

$$V_{\text{eff}}(x_1) \simeq \frac{\hbar^2}{4m} \left[ k_F^2(x_1)\text{sgn}(\lambda) - 2\lambda^2\theta(-\lambda) \right] . \tag{83}$$

In the case where $\lambda < 0$ we see that $V_{\text{eff}}(x_1)$ contains a term corresponding to the bound state energy, $E_b = -\hbar^2\lambda^2/2m$ of the state localized around the impurity. Again we see that repulsive impurities are repelled from dense regions and attractive impurities are attracted by dense regions, which clearly agrees with physical intuition.

**Link with the problem of mobile impurities.** The above results on the effective interaction potential felt by an impurity in an inhomogeneous system is to our knowledge new, as mentioned above the problem for a homogeneous system was discussed in [22, 23]. However a problem with a similar flavor, involving a *mobile* impurity, has been studied. McGuire [19, 20] considered the problem of $N$ identical spin-less fermions with no mutual

interactions. In this system, one introduces an additional particle, with coordinate $x_0$, which interacts with each of the $N$ fermions via a delta function potential. In its most general form we can consider the $N+1$ body Hamiltonian given by

$$H = \sum_{i=1}^{N} -\frac{\hbar^2}{2m}\frac{\partial^2}{\partial x_i^2} + V(x_i) + g\sum_{i=1}^{N}\delta(x_i - x_0) - \frac{\hbar^2}{2M}\frac{\partial^2}{\partial x_0^2} + \mathcal{V}(x_0), \tag{84}$$

where $M$ is the mass of the *impurity particle* and $\mathcal{V}(x)$ the effective potential it feels due to the trap. The problem examined by McGuire corresponds to identical masses, i.e. $M = m$ and to homogeneous system with $V(x) = 0$, hence $k_F$ (and thus the density) being constant. The change in the energy due to the additional particle is found to be

$$\Delta E(k_F, \lambda) = \frac{\hbar^2}{2\pi m}\left[(2k_F^2 + \frac{\lambda^2}{2})\tan^{-1}\left(\frac{\lambda}{2k_F}\right) + k_F\lambda - \frac{\pi\lambda^2}{4}\right]. \tag{85}$$

Interestingly, we note that this formula (85) is strikingly similar to the expression obtained here in the case of an immobile impurity and one can write

$$\Delta E(k_F, \lambda) = \frac{\hbar^2\lambda^2}{4\pi m}W\left(\frac{2\,k_F}{\lambda}\right), \tag{86}$$

with the same scaling function $W(\gamma)$ given in Eq. (14). Note that the problem considered in this paper corresponds to the limit $M \to \infty$ where the position $x_0$ is fixed.

More recently the problem introduced by McGuire was revisited in the presence of an external harmonic potential $V(x) = m\omega^2 x^2/2$, which is the same on both the fermions and the impurity particle. The effect of inhomogeneity was treated by combining McGuire's result with an LDA-like approximation, which turns out to be remarkably accurate even for systems with a small number of fermions [24].

## 5 Impurity far from the bulk and the filling transition

We consider an impurity of strength $g_1 = g$ placed at $x_1$ far from the bulk of the Fermi gas, where the condition in Eq. (29) holds and the density vanishes. One may ask the question whether an attractive impurity can pull fermions out of the bulk. In this region the kernel in the region of the point $x_1$ is given by

$$\Delta K_\mu(x_1 + z, x_1 + z') = \frac{1}{\pi}\text{Im}\int_{-\infty}^{\mu} d\mu'\, \Delta G_{\mu'}(x_1 + z, x_1 + z'), \tag{87}$$

where in the above integral $\mu' < \mu \ll V(x_1)$ (and thus the contour $\Gamma_2$ in Fig. 2 is the appropriate one to use). More precisely the condition in Eq. (29) holds in the above integral and so we can use Eq. (38) for the Green's function. When an impurity is placed at the point $x_1$ the induced change in the Green's function is

$$\Delta G_{\mu'}(x_1 + z, x_1 + z') = \frac{g\,G_{0\mu'}(x_1 + z, x_1)G_{0\mu'}(x_1, x_1 + z')}{1 - gG_{0\mu'}(x_1, x_1)} \tag{88}$$

$$= \frac{m\lambda}{\hbar^2}\frac{\exp(-(\kappa_{\mu'}(x_1) + i0^+)[|z| + |z'|])}{(\kappa_{\mu'}(x_1) + i0^+)([\kappa_{\mu'}(x_1) + i0^+] + \lambda)}, \tag{89}$$

where $\kappa_{\mu'}(x_1)$ is given by Eq. (39).

Now using $d\mu' = -\hbar^2\kappa d\kappa/m$ we find that the change in the kernel is

$$\Delta K_\mu(x_1 + z, x_1 + z') = \frac{\lambda}{\pi}\text{Im}\int_{\kappa_F(x_1)}^{\infty} d\kappa\frac{\exp(-(\kappa + i0^+)[|z| + |z'|])}{[\kappa + i0^+] + \lambda}. \tag{90}$$

Here and below we denote $\kappa_\mu(x_1) = \kappa_F(x_1)$. We now use the standard identity

$$-\frac{1}{\pi}\text{Im}\frac{1}{-\kappa - \lambda - i0^+} = -\delta(\kappa + \lambda), \tag{91}$$

which gives, for $\lambda > 0$,

$$\Delta K_\mu(x_1 + z, x_1 + z') = 0 , \tag{92}$$

as $\kappa_F(x_1) > 0$. Hence a repulsive impurity far from the bulk has no effect on the Fermi gas.

However for an attractive impurity, writing $\lambda = -\lambda_A$ with $\lambda_A > 0$ we find

$$\Delta K_\mu(x_1 + z, x_1 + z') = \lambda_A\theta(\lambda_A - \kappa_F(x_1))\exp(-\lambda_A[|z| + |z'|]). \tag{93}$$

The above result is easily interpreted physically. It can be written as

$$\Delta K_\mu(x_1 + z, x_1 + z') = \theta(\lambda_A - \kappa_F(x_1))\psi_b(z)\psi_b(z'), \tag{94}$$

where $\psi_b(z)$ is the bound state wave-function for a delta potential at $z = 0$ and in the absence of any other potential given in Eq. (65). The kernel well outside the bulk is thus generated by a single particle bound state. The fermion density at the position of the impurity is thus given by

$$\rho_\mu(x_1) = \lambda_A\theta(\lambda_A - \kappa_F(x_1)). \tag{95}$$

It exhibits a transition as a function of $\lambda_A$. It vanishes when $\lambda_A < \kappa_F(x_1)$ and is nonzero for $\lambda_A > \kappa_F(x_1)$. This corresponds to a *filling transition* of the bound state where the density exhibits a "jump" by $\kappa_F(x_1)$. The transition is sharp for large $\kappa_F(x_1)$. At smaller values of $\kappa_F(x_1)$, i.e. close to the edge, it is replaced by a smooth crossover, which is analysed in Section 7.

This transition can be interpreted by the following energy argument. The energy of this bound state in the presence of a local potential is given by

$$E_b^*(x_1) = -\frac{\lambda_A^2\hbar^2}{2m} + V(x_1) . \tag{96}$$

Hence this state is occupied if $E_b^* < \mu$, which corresponds to $\lambda_A > \kappa_F(x_1)$. In contrast, when $E_b^* > \mu$ this bound state energy level exceeds the Fermi energy and hence it remains unoccupied at zero temperature. We note that this type of transition is quite generic, and not specific to a delta-function impurity. For instance, it can occur in a more general context when there is a second additional potential well (i.e. a second minimum of the trapping potential). When the Fermi energy increases above this second minimum, a new disjoint interval arises in the support of the density. However the case of the delta potential yields a particularly simple tractable example, since it corresponds to a *rank one* perturbation.

A natural question to ask is whether there is an effective potential felt by the particle when it is outside the bulk. This potential can, again, be derived using the Hellmann-Feynman theorem (80) together with the expression for the density outside the bulk given in Eq. (95). We find

$$V_{\text{eff}}(x_1) = -\frac{\hbar^2}{2m}\theta(\lambda_A - \kappa_F(x_1))\left[\lambda_A^2 - \kappa_F(x_1)^2\right] . \tag{97}$$

Furthermore we note that using Eq. (39) we can write

$$V_{\text{eff}}(x_1) = \theta(\lambda_A - \kappa_F(x_1))\left[-\frac{\hbar^2}{2m}\lambda_A^2 + V(x_1) - \mu\right] . \tag{98}$$

Hence we see that for $\lambda_A > \kappa_F(x_1)$, the $x_1$-dependence of the effective potential $V_{\text{eff}}(x_1)$ is the same as the trapping potential $V(x_1)$, this is due to the fermion which forms a state bound about the impurity.

**An analogy in random matrix theory.** This *filling transition* is reminiscent of the Baik-Ben Arous-Péché (BBP) transition in random matrix theory [17, 18]. The BBP transition occurs when one considers a $N \times N$ random matrix $\mathcal{M}_0$, for instance from the GUE with a semi-circle density of support $[-\sqrt{2N}, \sqrt{2N}]$ at large $N$ (the Wigner sea), perturbed by a fixed ranked one matrix, i.e. when considering the matrix sum $\mathcal{M} = \mathcal{M}_0 + \sqrt{\frac{N}{2}}\gamma|e_1\rangle\langle e_1|$. For a weak perturbation $\gamma < 1$ the Wigner sea is essentially unchanged and the largest eigenvalue $\lambda_{\text{max}}$ of $\mathcal{M}$ behaves as in the absence of perturbation, i.e. $\lambda_{\text{max}} \simeq \sqrt{2N} + \frac{1}{\sqrt{2}N^{1/6}}\chi_2$ where $\chi_2 = O(1)$ fluctuates according to the GUE Tracy-Widom distribution. Above the threshold, for $\gamma > 1$, an outlier or spike eigenvalue detaches from the Wigner sea, and the largest eigenvalue now behaves as $\lambda_{\text{max}} \simeq \sqrt{\frac{N}{2}}(\gamma + \frac{1}{\gamma}) + \frac{1}{\sqrt{2}}\mathcal{N}(0, \sigma^2 = 1 - \frac{1}{\gamma^2})$ (i.e., with Gaussian fluctuations). The analogy with our quantum problem is suggested by the fact that (i) the joint PDF of the eigenvalues of $\mathcal{M}_0$ is identical to the joint PDF of the fermion positions in the ground state of the harmonic oscillator $V(x) = \frac{1}{2}x^2$ (ii) the perturbation is of rank one in each problem. It is then tempting to establish an analogy between the spike/outlier from the Wigner sea, and the fermion bound to the delta impurity outside the Fermi sea. A difference is that the analog of the typical value of $\lambda_{\text{max}}$ would be $x_1$, which in our problem is given. However, in both cases the order parameter of the strong coupling phase (i.e. $\gamma > 1$ in RMT or $\lambda_A > \kappa_F(x_1)$ in the fermion problem) is the overlap of the state of the system with the perturbation. Note that the BBP transition has a non-trivial critical regime when $\gamma - 1 = O(N^{-1/3})$, where $\lambda_{\text{max}}$ gradually leaves the edge of the spectrum. The critical region in our problem corresponds to the case where the impurity is placed near the edge studied below in Section 7.

## 6 Interaction between two impurities in the bulk

In this section we compute the effective interaction between two impurities in the bulk separated by a distance $r$. We place impurity 1 at $x_1$ and impurity 2 at $x_2$. We assume that $|x_2 - x_1| \ll \xi$ where the length $\xi$ is defined in Eq. (30) such that the trapping potential can be considered as constant. For two impurities we can still apply the Hellmann-Feymnam theorem, for example differentiating with respect to $\lambda_2$, which measures the interaction strength of impurity 2. Writing explicitly the dependence on the coupling constants $\lambda_1$ and $\lambda_2$, we obtain the analogue of Eq. (79) valid for two impurities

$$\frac{\partial \Omega(\mu, \lambda_1, \lambda_2)}{\partial \lambda_2} = \frac{\hbar^2}{m}\rho_\mu(x_2, \lambda_1, \lambda_2) , \tag{99}$$

where $\Omega(\mu, \lambda_1, \lambda_2)$ is the grand-potential of the system in the presence of the two impurities (it depends on both $x_1$ and $x_2$ but we omit the explicit dependence for notational simplicity). In Eq. (99), $\rho_\mu(x_2, \lambda_1, \lambda_2)$ denotes the density of the Fermi gas at the location of the second impurity. Let us define the effective interaction between the two particles as

$$V_{\text{int}}(r) = \Omega(\mu, \lambda_1, \lambda_2) - \Omega(\mu, \lambda_1, 0) - \Omega(\mu, 0, \lambda_2) + \Omega(\mu, 0, 0) , \tag{100}$$

which depends only on the distance $r = |x_2 - x_1|$ since the system is translationally invariant on scale of the order $O(\xi)$. The interaction potential $V_{\text{int}}(r)$ in Eq. (100) can be

written as

$$V_{\text{int}}(r) = \frac{\hbar^2}{m} \left[ \int_0^{\lambda_2} d\lambda_2' \ \rho_\mu(x_2, \lambda_1, \lambda_2') - \int_0^{\lambda_2} d\lambda_2' \ \rho_\mu(x_2, 0, \lambda_2') \right]. \tag{101}$$

Now using the representation of $\rho_\mu(x_2, \lambda_1, \lambda_2)$ in terms of the Green's function in Eq. (21) by setting $x = y = x_2$ we find

$$\rho_\mu(x_2, \lambda_1, \lambda_2) = K_\mu(x_2, x_2) = \frac{1}{\pi} \text{Im} \int_{-\infty}^\mu d\mu' G_{\mu'}(x_2, x_2). \tag{102}$$

We now use Eq. (51) which can be written as

$$G_{\mu'}(x_2, x_2) = -\frac{m}{\hbar^2} \frac{\partial}{\partial \lambda_2} \ln \left( \det[1 - \Lambda_g \mathcal{G}_0] \right), \tag{103}$$

thus facilitating the integration with respect to $\lambda_2'$ in Eq. (101). This then yields

$$V_{\text{int}}(r) = -\frac{1}{\pi} \text{Im} \int_{-\infty}^\mu d\mu' \ \Big[ \ln \big( (1 - g_1 G_{0\mu'}(x_1, x_1))(1 - g_2 G_{0\mu'}(x_2, x_2)) - g_1 g_2 G_{0\mu'}^2(x_1, x_2) \big)$$

$$- \ln \big( 1 - g_1 G_{0\mu'}(x_1, x_1) \big) - \ln \big( 1 - g_2 G_{0\mu'}(x_2, x_2) \big) \Big]. \tag{104}$$

Using the fact that $G_{0\mu}(x, y) = G_{0\mu}(x - y)$, we get

$$V_{\text{int}}(r) = -\frac{1}{\pi} \text{Im} \int_{-\infty}^\mu d\mu' \ \ln \left( 1 - \frac{g_1 g_2 G_{0\mu'}^2(r)}{(1 - g_1 G_{0\mu'}(0))(1 - g_2 G_{0\mu'}(0))} \right), \tag{105}$$

where we recall that $r = |x_2 - x_1|$. Now using the expression for the Green's function in Eq. (25), valid in the bulk, and again changing the integration variable to $k$ where $\mu' = \hbar^2 k^2 / 2m$, we find

$$V_{\text{int}}(r) = -\frac{\hbar^2}{\pi m} \text{Im} \int_{\Gamma_2'} dk \, k \ln \left( 1 + \frac{\lambda_1 \lambda_2}{[k - i\lambda_1][k - i\lambda_2]} \exp(-2ikr) \right), \tag{106}$$

where the contour $\Gamma_2'$ is shown in Fig. 3. As $\lambda_1$ and $\lambda_2$ are real, there are no poles inside the region enclosed by the contours $\Gamma_2'$, $\Gamma_4$ and $\Gamma_3''$ shown in Fig. 3. Therefore Cauchy's theorem tells us that the contour integral around this region is identically zero. Using further the fact that the integrand vanishes on the contour $\Gamma_3''$ as its radius is extended to $\infty$ we obtain

$$V_{\text{int}}(r) = \frac{\hbar^2}{\pi m} \text{Im} \int_{\Gamma_4} dk \, k \ln \left( 1 + \frac{\lambda_1 \lambda_2 \exp(-2ikr)}{[k - i\lambda_1][k - i\lambda_2]} \right). \tag{107}$$

Making the substitution $k = k_F - i\kappa$ where $k_F = k_F(x_1) \approx k_F(x_2)$, we find the exact result

$$V_{\text{int}}(r) = -\frac{\hbar^2}{\pi m} \text{Im} \, i \int_0^\infty d\kappa (k_F - i\kappa) \ln \left( 1 + \frac{\lambda_1 \lambda_2 \exp(-2ik_F r - 2\kappa r)}{[k_F - i\kappa - i\lambda_1][k_F - i\kappa - i\lambda_2]} \right). \tag{108}$$

Writing $\lambda_i = k_F \gamma_i$ and making the change of variable $\kappa = k_F u$ we find the interaction energy, with its dependence on the physical parameters $r$, $k_F$, $\gamma_1$ and $\gamma_2$ made explicit, is given by

$$
\begin{aligned}
V_{\text{int}}(r, k_F, \gamma_1, \gamma_2) &= -\frac{\hbar^2 k_F^2}{\pi m} \text{Im} \, i \int_0^\infty du \, (1 - iu) \ln \left( 1 + \frac{\gamma_1 \gamma_2 \exp(-2ik_F r - 2k_F ur)}{[1 - iu - i\gamma_1][1 - iu - i\gamma_2]} \right) \\
&= -\frac{\hbar^2 k_F^2}{\pi m} \text{Re} \int_0^\infty du \, (1 - iu) \ln \left( 1 + \frac{\gamma_1 \gamma_2 \exp(-2k_F r(i + u))}{[1 - iu - i\gamma_1][1 - iu - i\gamma_2]} \right).
\end{aligned}
\tag{109}
$$

For $k_F r \gg 1$ one can expand the integrand about $u = 0$. The corrections due to terms of order $u$ are of order $1/k_F r$ (see Eq. (112) below). We thus find for $r$ large

$$V_{\text{int}}(r, k_F, \gamma_1, \gamma_2) \simeq -\frac{\hbar^2 k_F^2}{\pi m}\text{Re}\int_0^\infty du \ln\left(1 + \frac{\gamma_1\gamma_2\exp(-2k_F r(i+u))}{[1-i\gamma_1][1-i\gamma_2]}\right). \quad (110)$$

This is exactly the result obtained in [13, 14] via a field theoretic method based on the summation of Matsubara frequencies, which becomes a continuous integral at zero temperature. In this large distance approximation the integral can be evaluated to give

$$V_{\text{int}}(r, k_F, \gamma_1, \gamma_2) \simeq \frac{\hbar^2 k_F}{2r\pi m}\text{Re}\,\text{Li}_2\left(-\frac{\gamma_1\gamma_2\exp(-2ik_F r)}{[1-i\gamma_1][1-i\gamma_2]}\right), \quad (111)$$

where $\text{Li}_2$ is the di-logarithm function [28].

The interaction potential can then be written in terms of the Fermi energy $E_F = \frac{\hbar^2 k_F^2}{2m}$ and the variable $\zeta = k_F r$ to give

$$V_{\text{int}}(r, k_F, \gamma_1, \gamma_2) = -\frac{2E_F}{\pi\zeta}\text{Re}\int_0^\infty ds\left(1-i\frac{s}{\zeta}\right)\ln\left(1 + \frac{\gamma_1\gamma_2\exp(-2\zeta i - 2s))}{[1-i\frac{s}{\zeta}-i\gamma_1][1-i\frac{s}{\zeta}-i\gamma_2]}\right), \quad (112)$$

and with the asymptotic form for large $\zeta$ given by

$$V_{\text{int}}(r, k_F, \gamma_1, \gamma_2) \simeq \frac{E_F}{\pi\zeta}\text{Re}\,\text{Li}_2\left(-\frac{\gamma_1\gamma_2}{[1-i\gamma_1][1-i\gamma_2]}\exp(-2i\zeta)\right). \quad (113)$$

In Fig. 6a we show the interaction energy in units of $E_F$, $U(\zeta, \gamma_1, \gamma_2) = V_{\text{int}}(\zeta, k_F, \gamma_1, \gamma_2,)/E_F$, for the exact result Eq. (112) for two repulsive impurities with $(\gamma_1, \gamma_2) = (1, 1)$, as a function of $\zeta$, along with the corresponding large distance approximation of Eq. (113). As was the case for Friedel oscillations the asymptotic result is accurate for $\zeta > 3$ but diverges towards $-\infty$ as $\zeta \to 0$. The potential oscillates in a manner reminiscent of the Friedel oscillations, exhibiting local minima, but is attractive for $\zeta < 1$. In Fig. 6b we show the corresponding result for two attractive impurities $(\gamma_1, \gamma_2) = (-1, -1)$, again the potential oscillates and presents local minima. A sharp barrier appears at $\zeta \sim 1/2$, but a deep minimum is formed for small $\zeta$. In Fig. 6c we show the case for an attractive and repulsive impurity with $(\gamma_1, \gamma_2) = (1, -1)$. Here, in contrast to the case of impurities of the same type, the short distance behavior of the potential is repulsive.

# 7   Impurity near the edge

In this section we investigate the effect of adding a delta function impurity near the edge at $x = x_e$ such that $\mu = V(x_e)$. In this region, in the absence of the impurity the Green's function satisfies the scaling form given in Eq. (33). The width of this region is given by $w_e$ and the corresponding energy scale is denoted by $\alpha_e$, both displayed in Eq. (34). Consider now an impurity located at $x = x_1 = x_e + z_0$ where $z_0$ will be of the same order as $w_e$. Substituting the scaling form from (33) into (52), we obtain the change in the Green's function due to the impurity as

$$\Delta G_{\mu'}(x_1 + z, x_1 + z') = \frac{\lambda^*}{\alpha_e w_e}\frac{g_e\left(\frac{z+z_0}{w_e} + \frac{\mu-\mu'}{\alpha}, \frac{z_0}{w_e} + \frac{\mu-\mu'}{\alpha_e}\right)g_e\left(\frac{z_0}{w_e} + \frac{\mu-\mu'}{\alpha}, \frac{z'+z_0}{w_e} + \frac{\mu-\mu'}{\alpha_e},\right)}{1 - \lambda^* g_e\left(\frac{z_0}{w_e} + \frac{\mu-\mu'}{\alpha_e}, \frac{z_0}{w_e} + \frac{\mu-\mu'}{\alpha_e}\right)},$$
$$(114)$$

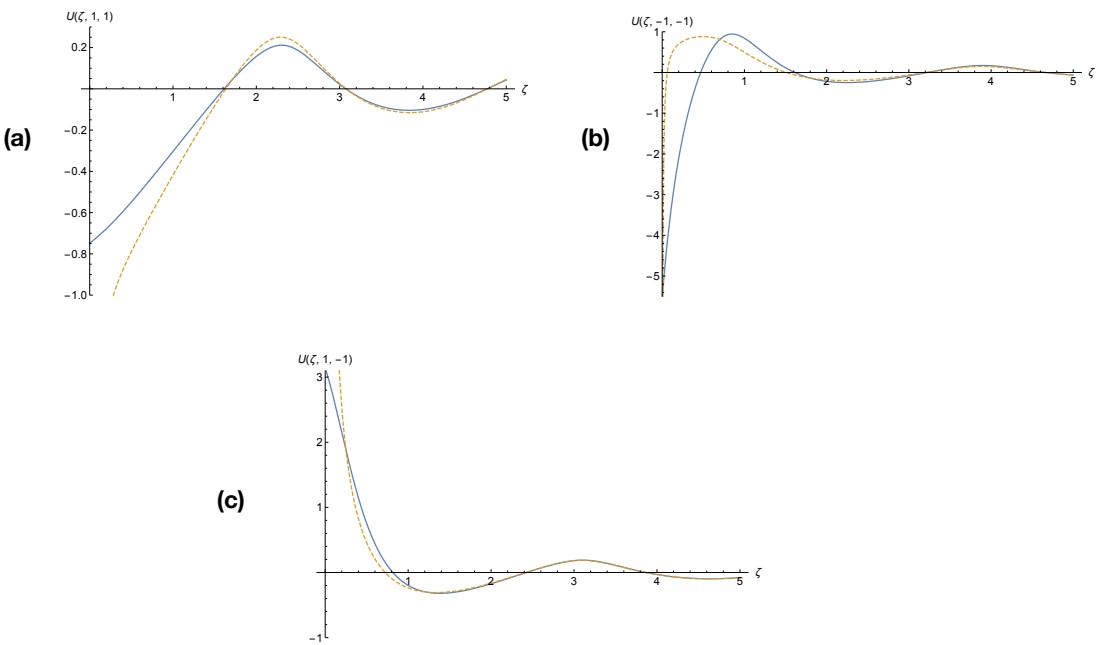

Figure 6: Effective interaction $U(z, \gamma_1, \gamma_2)$ (in units of the Fermi energy) between impurities, solid lines exact interaction given by Eq. (112) and dashed lines asymptotic large distance approximation Eq. (113). Shown in (a), (b) and (c) impurities with interactions strengths $(\gamma_1, \gamma_2) = (1, 1)$, $(-1, -1)$ and $(1, -1)$ respectively.

where $g_e(\zeta, \zeta')$ is given in (35) and

$$\lambda^* = \frac{\hbar^2 \lambda}{m \alpha_e w_e} = 2\lambda w_e \tag{115}$$

is a dimensionless measure of the impurity strength in the edge region.

The kernel $K_{0\mu}$ at the edge in the absence of impurity is given by,

$$K_{0\mu}(x, y) = \frac{1}{w_e} K_{\mathrm{Ai}}\left(\frac{x - x_e}{w_e}, \frac{y - x_e}{w_e}\right) \quad , \quad K_{\mathrm{Ai}}(a, b) = \int_0^\infty du \, \mathrm{Ai}(a + u)\mathrm{Ai}(b + u) \tag{116}$$

in terms of the Airy kernel $K_{\mathrm{Ai}}$. The change in the kernel, $\Delta K_\mu = K_\mu - K_{0\mu}$, is obtained from Eq. (24) by integrating over $\mu'$ between $\mu$ and $+\infty$, making the change of variables $\frac{z_0}{w_e} + \frac{\mu - \mu'}{\alpha_e} = -u$ and setting $z_0 = c w_e$ then gives

$$\Delta K_\mu(x_1 + a w_e, x_1 + b w_e) = -\frac{\lambda^*}{\pi w_e} \int_{-c}^\infty du \, \mathrm{Im} \, \frac{g_e(a - u, -u)g_e(-u, b - u)}{1 - \lambda^* g_e(-u, -u)} \quad , \tag{117}$$

where function $g_e$ is given in (35) and the dimensionless number $c$ given by

$$c = \frac{z_0}{w_e} = \frac{x_1 - x_e}{w_e} \, , \tag{118}$$

measures the relative position of the impurity compared to the edge. The above integral has an integrand which oscillates and decays like $1/u$ for large $u$ and it cannot, at least in any obvious sense, be evaluated analytically. However if we use Eq. (23) we find the alternative expression

$$\Delta K_\mu(x_1 + a w_e, x_1 + b w_e) = \frac{\lambda^*}{\pi w_e} \int_c^\infty du \, \mathrm{Im} \, \frac{g_e(a + u, u)g_e(u, b + u)}{1 - \lambda^* g_e(u, u)} \quad , \quad c = \frac{x_1 - x_e}{w_e}, \tag{119}$$

which converges quickly for $u \to +\infty$ allowing an efficient numerical integration (see below).

## 7.1 Density at the edge

Of particular interest is how the density is modified by the presence of a delta function at the edge. The average density of the Fermi gas around the impurity, can now be obtained by setting coinciding points in the total kernel which, in terms of the scaled position $a$ measured from the position of the delta impurity, leads to

$$\rho_\mu(x) = \frac{n(a, c, \lambda^*)}{w_e}, \quad , \quad a = \frac{x - x_1}{w_e} \tag{120}$$

where

$$n(a, c, \lambda^*) = \int_c^\infty du \left[ \mathrm{Ai}(u + a)^2 + \frac{\lambda^*}{\pi} \mathrm{Im} \frac{g_e(a + u, u)^2}{1 - \lambda^* g_e(u, u)} \right] \quad , \quad c = \frac{x_1 - x_e}{w_e} \tag{121}$$

and $c$ is the scaled relative position of the delta impurity with respect to the edge. In the above, when $\lambda^* = 0$ we recover the usual edge density of the Airy gas which, in random matrix theory, corresponds to the eigenvalue density for the Gaussian Unitary Ensemble at the edge where the Wigner semi-circle law vanishes [31]. Using (35) the integrand in the second term can be written more explicitly as

$$D(a, u) = \frac{\lambda^*}{\pi} \mathrm{Im} \frac{g_e(a + u, u)^2}{1 - \lambda^* g_e(u, u)} \tag{122}$$

$$= \begin{cases} -\pi \lambda^* \mathrm{Ai}(u + a)^2 \mathrm{Ai}(u) \frac{\pi \lambda^* \mathrm{Ai}(u)^3 + \pi \lambda^* \mathrm{Ai}(u) \mathrm{Bi}(u)^2 + 2 \mathrm{Bi}(u)}{(1 + \pi \lambda^* \mathrm{Ai}(u) \mathrm{Bi}(u))^2 + (\pi \lambda^*)^2 \mathrm{Ai}(u)^4} \quad , & a > 0 \\ \\ -\pi \lambda^* \mathrm{Ai}(u)^2 \frac{\pi \lambda^* \mathrm{Ai}(u)^2 (\mathrm{Bi}(u+a)^2 - \mathrm{Ai}(u+a)^2) + 2 \mathrm{Ai}(u+a) \mathrm{Bi}(u+a)(1 + \pi \lambda^* \mathrm{Ai}(u) \mathrm{Bi}(u))}{(1 + \pi \lambda^* \mathrm{Ai}(u) \mathrm{Bi}(u))^2 + (\pi \lambda^*)^2 \mathrm{Ai}(u)^4} \quad , & a < 0 \,. \end{cases} \tag{123}$$

Let us recall the asymptotic behavior of the Airy functions. For $u \to +\infty$ one has

$$\mathrm{Ai}(u) \simeq \frac{\exp(-\frac{2}{3} u^{\frac{3}{2}})}{2 \sqrt{\pi} u^{\frac{1}{4}}} \,, \; \mathrm{Bi}(u) \sim \frac{\exp(\frac{2}{3} u^{\frac{3}{2}})}{\sqrt{\pi} u^{\frac{1}{4}}}, \tag{124}$$

which implies that $\mathrm{Ai}(u) \mathrm{Bi}(u) \simeq \frac{1}{2\pi \sqrt{u}}$, and so we see that for $u \to +\infty$ the r.h.s. of (123) behaves as $\simeq -2\pi \lambda^* \mathrm{Ai}(u)^3 \mathrm{Bi}(u)$ and thus decays very quickly. On the other side, for $u \to -\infty$ both $\mathrm{Ai}(u)$ and $\mathrm{Bi}(u)$ decay as $1/|u|^{1/4}$ with oscillating prefactors. Hence it seems that the integral can be easily evaluated numerically (apart from a subtlety arising from the denominator for large negative $\lambda^*$ see below). Note also that as $a \to \pm\infty$ the change in the density due to the impurity decays to zero.

In Fig. 7 we have plotted $n(a, 0, \lambda^*)$ as a function of $a$ for the values $\lambda^* = 0$, the case where there is no impurity at the edge, and for value $\lambda^* = 1$ (repulsive impurity) and the value $\lambda^* = -1$ (attractive impurity) for an impurity placed exactly at the edge $c = 0$. We see that in all cases, the density oscillates in the region to the left of the edge (as one moves towards the bulk) but decays monotonically to the right as one moves away from the bulk. The presence of a repulsive impurity decreases the density, as expected, and induces a small phase shift in the oscillations to the left. However the attractive impurity increases the density at the original edge and leads to a larger density of fermions to the right, again with monotonic decay. In addition, the oscillations in the density experience a substantial phase shift with respect the the case of no impurity and a repulsive impurity. The presence of an impurity introduces a discontinuity in the derivative of the density

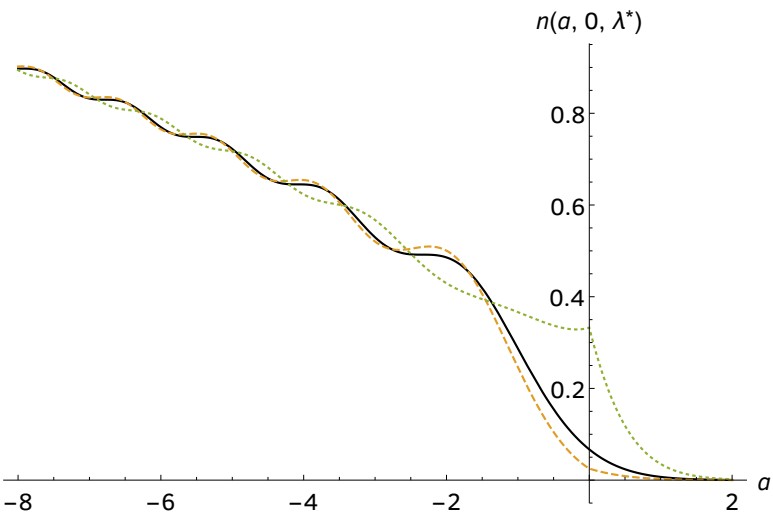

Figure 7: The rescaled density $n(a, 0, \lambda^*)$ as a function of the rescaled distance $a$, for a delta function interaction placed at the edge of a trap $c = 0$ on top of a potential which is locally linear at the edge. In black solid is the case where $\lambda^* = 0$, that is to say no perturbation. In dashed is shown the case where $\lambda^* = 1$ (repulsive impurity) while the case $\lambda = -1$ (attractive impurity) is shown by the dotted line.

at $x_1$, i.e. at $a = 0$. Similar effects are seen when the impurity is not placed exactly at the edge $x_1 \neq x_e$. For an attractive impurity placed on the right of the edge, the density increases around the impurity.

*Filling transition* **of an attractive impurity located far from the edge**. Here we examine, within the edge region, the filling transition already discussed in Section 5 in the context of the bulk. Clearly if the delta impurity is placed far to the right of the edge, i.e. $c = \frac{x_1 - x_e}{w_e} \gg 1$, the local potential at the position of the impurity, $V(x_1)$, is large compared to the value $V(x_e)$ at the edge. Being well above the Fermi sea it should play no role in the Fermi gas, unless the amplitude of the delta impurity, $\lambda = -\lambda_A < 0$, is tuned to be sufficiently attractive. Indeed, in that case we can revisit the qualitative argument given in Section 5. An attractive delta impurity in a uniform potential produces a bound state with a binding energy $E_b = -\frac{\hbar^2}{2m}\lambda_A^2 = m\alpha_e^2 w_e^2 \lambda_A^{*2}/(2\hbar^2)$, hence its total energy is $V(x_1) + E_b$. We can now surmise that when this energy is lowered below the Fermi energy $\mu = V(x_e)$, this bound state should be filled and be part of the ground state of the Fermi gas. If one equates this binding energy with the energy shift $V(x_1) - V(x_e) = V'(x_1)(x_1 - x_e) = V'(x_1)w_e c$ by linearizing the potential near the edge, one finds that the transition should occur at

$$\lambda_A^* \simeq 2\sqrt{c} \qquad (125)$$

for large $c \gg 1$. It turns out that the estimate (125) is quantitatively correct, as we now show.

To see this we return to the formula (121) and (123) for the density around the impurity and recall from (35) that $g_e(u, u) = -\pi \text{Ai}(u)[-i\text{Ai}(u) + \text{Bi}(u)]$. If $c \gg 1$, then the integration region is $u > c \gg 1$ and we can use the asymptotics (124), and $\text{Ai}(u)\text{Bi}(u) \simeq \frac{1}{2\pi\sqrt{u}}$

for $u \gg 1$. We see that the denominator in (123) becomes at large $u$

$$\frac{1}{1 - \lambda^* g_e(u,u)} \simeq \frac{1}{1 + \frac{\lambda^*}{2u^{\frac{1}{2}}} - i\lambda^* \frac{\exp(-\frac{4}{3}u^{\frac{3}{2}})}{4u^{\frac{1}{2}}}}, \tag{126}$$

We see that the real part of the denominator vanishes at $u = u_c = \lambda_A^{*2}/4$ when $\lambda^* = -\lambda_A^* < 0$, i.e. for an attractive impurity. To study the transition, from (125) we should consider $\lambda_A$ large, hence $u_c \gg 1$. In this case since the imaginary part is very small we can make the approximation

$$\frac{1}{1 - \lambda^* g_e(u,u)} \simeq P\frac{1}{1 - \frac{\lambda_A^*}{2u^{\frac{1}{2}}}} - i\pi\delta\left(1 - \frac{\lambda_A^*}{2u^{\frac{1}{2}}}\right) = P\frac{1}{1 - \frac{\lambda_A^*}{2u^{\frac{1}{2}}}} - \frac{i\pi\lambda_A^{*2}}{2}\delta(u - u_c), \tag{127}$$

where $P$ denotes the Cauchy principle part. This means that the local density of states has a sharp resonance at $u = u_c$. Inserting into (123) and (121) we see that the term which converges to a delta function gives a contribution

$$D_\delta(a,u) = -\frac{\lambda^3}{2}(\text{Re}[g_e(a + u_c, u_c)])^2 \simeq \delta(u - u_c)\frac{\lambda_A^*}{2}\exp(-\lambda_A^*|a|). \tag{128}$$

Using $\text{Re}[g_e(a + u_c, u_c)] \simeq \frac{1}{2\sqrt{u_c}}e^{-2a\sqrt{u_c}}$ for large $u_c$, we find that the corresponding contribution to the local density reads

$$n_\delta(a,c) \simeq \theta(u_c - c)\frac{\lambda_A^*}{2}\exp(-\lambda_A^*|a|). \tag{129}$$

This contribution was obtained under the assumption that $u_c \gg 1$. We see that it is non zero if $u_c > c$, that is for $\lambda_A^* > 2\sqrt{c}$, exactly the same condition as (125). In that case the above contribution (129) corresponds precisely to a total of one particle, since its integral over $a$ is equal to 1. Hence for $\lambda_A^* > 2\sqrt{c}$ there is a local density peak corresponding to a total of one fermion. When $\lambda_A^* < 2\sqrt{c}$ this extra fermion is no longer present.

We see that this transition is very sharp for $c \gg 1$ and thus coincides with the transition discussed in section 5. One can perform a slightly more precise estimate of the above formula (121) and (123) for the density near the impurity and obtain for $a\lambda_A^* = O(1)$ and $c \gg 1$, the Lorentzian dependence in the impurity position $c$ near the transition at $c = u_c$

$$n(a,c,-\lambda_A^*) \simeq \frac{\lambda_A^*}{2}\exp(-\lambda_A^*|a|)\int_c^{+\infty} du\frac{1}{\pi}\frac{\eta}{(u - u_c)^2 + \eta^2}, \tag{130}$$

where

$$\eta = \frac{\pi(\lambda_A^*)^3}{2}\text{Ai}(u_c)^2 \quad, \quad u_c = \lambda_A^2/4. \tag{131}$$

Hence the width $\eta$ is exponentially small, i.e. $\eta \simeq e^{-\frac{4}{3}u_c^{3/2}}$.

Note that from the denominators in (123) we see that the effect described above persists for smaller values of $c = O(1)$, at the location $u_c$ of the root of $\text{Ai}(u_c)\text{Bi}(u_c) = -\frac{1}{\pi\lambda_A}$. However it is broader if both $c, u_c = O(1)$. Hence it is a crossover for $c = O(1)$ and becomes a sharp transition as $c \to +\infty$.

It is important to also compute the fermion density at the position of the impurity, e.g. to derive the effective potential in the next section. It is obtained by setting $x_1 = x$ and thus $a = 0$ in (132). Recalling that the first term is the imaginary part of $\frac{1}{\pi}g_e(u,u)$ we see that the formula simplifies into

$$\rho_\mu(x_1) = \frac{1}{w_e}\frac{1}{\pi}\text{Im}\int_c^\infty du\,\frac{g_e(u,u)}{1 - \lambda^* g_e(u,u)} = \frac{1}{w_e\lambda^*}\frac{1}{\pi}\text{Im}\int_c^\infty du\,\frac{1}{1 - \lambda^* g_e(u,u)}. \tag{132}$$

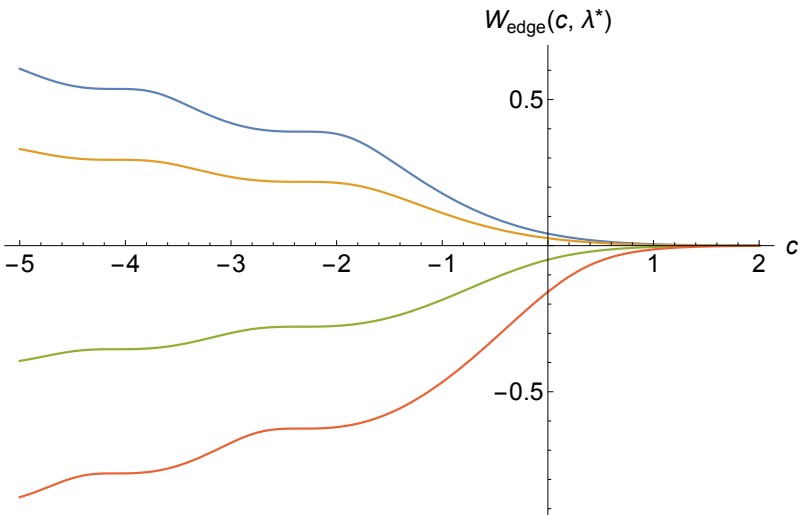

Figure 8: The scaled potential $W_{\text{edge}}(c, \lambda^*)$ felt by an impurity at the edge as a function of the distance $c$ from the edge measured in units of $w_e$ as give in Eq. (135). Shown from top to bottom is the potential for $\lambda^* = 1,\ 0.5,\ -0.5,\ -1$.

In the limit where $c = (x_1 - x_e)/w_e \to -\infty$, i.e. when the position of the impurity enters the bulk, one can easily check, using the explicit expression of $w_e$ in Eq. (34) and of $\lambda^*$ in Eq. (115), that $\rho_\mu(x_1) \simeq k_F(x_1)/\pi$, independently of the sign of $\lambda$, i.e. both for a repulsive and an attractive impurity. This behavior matches perfectly with the behavior found in the bulk in Eqs. (63) and (66) in the limit $|\lambda| \ll k_F$. This is expected since that the edge scaling form in Eq. (132) holds for finite $\lambda^*$, which implies $\lambda \simeq 1/w_e$ [see Eq. (115)], and thus $|\lambda| \ll k_F$ (since $w_e \gg 1/k_F$ for large $\mu$).

### 7.2 Effective potential at the edge

We now calculate the effective potential felt by an impurity in the edge region, as defined in (74). We can again use the Hellmann-Feynman theorem as in (80) which requires the density at the location of the impurity as given in (132). Since this density is expressed in terms of $\lambda^*$ it is convenient to write the Hellmann-Feynman formula in terms of $\lambda^*$ using (115). It reads

$$\frac{\partial V_{\text{eff}}(x_1, \lambda)}{\partial \lambda} = \frac{\hbar^2}{m \alpha_e w_e} \frac{\partial V_{\text{eff}}(x_1, \lambda)}{\partial \lambda^*} = \frac{\hbar^2}{m} \rho_\mu(x_1) \tag{133}$$

This equation can then easily be integrated with respect to $\lambda^*$ using Eq. (132) to obtain

$$V_{\text{eff}}(x_1, \lambda) = \alpha_e W_{\text{edge}}\left(\frac{x_1 - x_e}{w_e}, \lambda^*\right), \tag{134}$$

with

$$W_{\text{edge}}(c, \lambda^*) = \frac{1}{\pi} \int_c^\infty du \tan^{-1}\left(\frac{\lambda^* \pi \text{Ai}(u)^2}{1 + \lambda^* \pi \text{Ai}(u)\text{Bi}(u)}\right). \tag{135}$$

The function $W_{\text{edge}}(c, \lambda^*)$ can be evaluated numerically and is plotted in Fig. 8 as a function of $c$ for several values of $\lambda^*$. At large negative values of $\lambda^*$ the numerical evaluation becomes difficult, presumably due to the formation of a bound state about the impurity to the left of the edge. A more detailed analysis of this regime would be interesting to pursue and we leave this for future work. One can also verify, numerically, that Eq. (135) matches with the bulk form given in Eq. (13) as it should (see the discussion below Eq. (132)),

although an analytic demonstration is not obvious given the highly oscillatory nature of the integrand.

## 8 Discussion

In this paper we studied non interacting fermions in a trap at zero temperature, in the presence of a singular potential created by delta function impurities. The presence of these impurities changes the density of the Fermi gas around the impurities. For a single impurity the change in the density profile has been studied using a number of techniques from condensed matter physics. These methods have allowed the characterization of the density at distances far from the impurity, which shows the celebrated Friedel oscillations. In this paper, using a Green's function method developed in our previous work we have computed the exact form of the density at all distances from the impurity. Furthermore our method goes beyond the one point function and also allowed to obtain the quantum correlations by computing the central object known as the kernel. In addition this allowed us to compute the effective potential felt by the impurity.

We have shown how the behavior of the density and of the effective potential changes as one moves the impurity from the bulk of the Fermi gas to the edge created by the confining potential. We also unveiled an interesting "filling transition" which occurs when the impurity is moved outside of the support of the density of the Fermi gas. All these results are exact and non-perturbative in the strength of the impurity.

In addition when a pair of impurities is placed in the bulk of the Fermi gas at a distance $r$ from each other, the fermion background gives rise to an effective interaction $V_{\text{int}}(r)$ between them, much like the Casimir effect in quantum electrodynamics. We have calculated exactly this effective interaction $V_{\text{int}}(r)$ at all distances, and our formula agrees with previous results known only for large distances.

In this paper all calculations in the presence of impurities are performed in the ensemble where the Fermi energy $\mu$ is fixed, and the system is in contact with a reservoir, so that the number of fermions can vary. This corresponds to the grand-canonical ensemble (here at zero temperature). In the Appendix B we briefly discuss the possible differences which may appear if instead one works in the canonical ensemble where the number of fermions is fixed (isolated system) as the impurity strength and position may vary.

We have focused here on the zero temperature limit, however it is important to derive the results at finite temperature since experiments are usually conducted at finite temperature. Indeed, the results derived here can be extended to finite temperature $T$ in a straightforward way. As shown in [5, 32] the kernel at finite temperature in the grand canonical ensemble at chemical potential $\tilde{\mu}$ can be obtained from the zero temperature kernel, a relation which in the present framework can be written as

$$K_{\tilde{\mu}}(x, y) = \frac{1}{\pi} \int d\mu' \frac{1}{1 + e^{\beta(\mu' - \tilde{\mu})}} \text{Im} \, G_{\mu'}(x, y) \,, \tag{136}$$

with $\beta = 1/(k_B T)$. Using this expression, integral formulas can be obtained for all of the quantities studied in this paper. It would be challenging to analyse these formula in the future.

Another line of investigation would be to study the Wigner function [35, 36] in the neighborhood of impurities both in the bulk and at the edge [37]. As well as being interesting in its own right, this might be a first step to understand the dynamics of systems in the presence of impurities as the Wigner function turns out to be a useful tool in the context of dynamics [38, 39].

Finally, another interesting problem for further investigation is the question about a mobile impurity in a Fermi gas. This problem has been studied in several works, notably by McGuire [19, 20]. It would be interesting to see if one could develop a general theory which extrapolates between the static impurity case studied here and the mobile impurity problem, which could explain the similarities between the two cases which we unveiled in Eq. (86).

## Acknowledgements

We warmly thank N. R. Smith for insightful discussions and ongoing collaborations on related topics as well as useful comments on the manuscript. We are grateful to Z. Ristivojevic for pointing out useful references. This research was supported by ANR grant ANR-17-CE30-0027-01 RaMaTraF.

## A  Comparison with the results of Ref. [12]

In this appendix, we compare our exact result for the density in the presence of a delta-function impurity in the bulk, given in Eq. (60), to the formula obtained in Ref. [12] by a quite different method. For this purpose, it is convenient to start from the formula given in Eq. (57). This formula can also be represented using the contour $\Gamma'_2$ (see Fig. 3) which yields

$$\Delta K_\mu(x,y) = -\frac{\lambda}{\pi}\mathrm{Im}\int_{\Gamma'_2} dk \frac{\exp(-ik[|x|+|y|])}{k-i\lambda} \ . \tag{137}$$

Note however that this representation is *only valid for uniform systems*, as it assumed that the bulk approximation for the Green's function is valid for small $k$, which in general is not.

**The case $\lambda > 0$:** in this case there is no bound state and there is no contribution to $\Delta K_\mu(x,y)$ in Eq. (137) coming from the part of $\Gamma'_2$ along the negative imaginary axis. Assuming that the system is homogeneous, taking the imaginary part in Eq. (137) we find

$$\Delta K_\mu(x,y) = \frac{1}{\pi}\int_0^{k_F} dk \ \frac{k\lambda\sin(k[|x|+|y|]) - \lambda^2\cos(k[|x|+|y|])}{k^2+\lambda^2}. \tag{138}$$

Using this representation when one sets $x = y$ we obtain the formula of [12] for the change in the density - however we note that there is a factor of 2 difference as in [12] spin 1/2 fermions were treated.

**The case $\lambda < 0$:** When $\lambda < 0$ the integral over the contour $\Gamma'_2$ in (137) picks up a *half pole* contribution at $k = i\lambda$, we thus find

$$\Delta K_\mu(x,y) = -\frac{\lambda}{\pi}\mathrm{Im}\int_0^{k_F} dk \frac{\exp(-ik[|x|+|y|])}{k-i\lambda} - \lambda\theta(-\lambda)\exp(\lambda[|x|+|y|]) \ , \tag{139}$$

where the last term comes from the negative imaginary axis and corresponds to a bound state. The contribution from this bound state was in fact overlooked in Ref. [12]. In fact the formula Eq. (138) was computed in [12] via a direct summation of eigenfunctions, however when $\lambda < 0$ this formula misses the bound state which is introduced by an attractive impurity. Note however that the omission of this bound state does not affect the behavior

at large distance from the impurity of the Friedel oscillations since the contribution of the bound state to the density decays exponentially at large distance.

**The small $\lambda$ limit:** We note that taking the small $\lambda$ limit in Eq. (60) gives, to order $O(\lambda)$

$$\Delta\rho_\mu(x) \approx \frac{\lambda}{\pi}\text{Im } \text{E}_1(2ik_F|x|) = \frac{\lambda}{\pi}\text{si}(2k_F|x|), \tag{140}$$

where

$$\text{si}(z) = -\int_z^\infty dt\frac{\sin(t)}{t}, \tag{141}$$

is the sine integral [28]. This matches perfectly with the linear response formula derived in [12].

# B  Canonical ensemble

We discuss here how one would approach the problem of adding impurities in the canonical ensemble where the number of particles if fixed and equal to $N$. Consider for example adding one impurity of strength $\lambda$. The number of single particle energy levels below the Fermi energy $\mu$ (i.e. the integrated density of states) is given by

$$N(\mu, \lambda) = \sum_k \theta(\mu - \epsilon_k(\lambda)) \tag{142}$$

which is a function of $\lambda$. In order that $N$ be fixed $\mu$ must be a function of $\lambda$, $\mu(\lambda)$ such that

$$N(\mu(\lambda), \lambda) = \int dx \ \rho_{\mu(\lambda)}(x, \lambda) = N \tag{143}$$

Let us define the change, due to the introduction of the impurity, in the integrated density of states as

$$\Delta N(\mu, \lambda) = N(\mu, \lambda) - N(\mu, 0). \tag{144}$$

Given that the energy change is of order 1 and that the perturbation in the density is local it is clear that $\Delta N(\mu, \lambda)$ is also of order 1 [33]. Denoting $\Delta\mu = \mu(\lambda) - \mu(0)$ with $\mu(0) = \mu$ we can rewrite (143) as

$$N(\mu + \Delta\mu, 0) + \Delta N(\mu + \Delta\mu, \lambda) = N = N(\mu, 0), \tag{145}$$

where $\Delta\mu$ is the shift in the Fermi energy due to the impurity. To analyse what happens in the canonical ensemble one must carry out the computations in this paper at chemical potential $\mu + \Delta\mu$, so the total fermion number is fixed upon adding the impurity. However, if $\Delta\mu$ is zero, then the results in this paper can simply be applied to the canonical ensemble.

A first example of where the canonical and grand canonical ensembles are equivalent is in a bulk system of volume $V$ where one has a total particle number $N = N(\mu, 0) = \frac{k_F(\mu)V}{\pi}$. Now using Eq. (26) for a bulk system, $k_F(\mu) = \sqrt{2m\mu}/\hbar$, we see that $\mu = \hbar^2\pi^2\frac{N^2}{2mV^2}$ and so $\Delta\mu \simeq -\hbar^2\pi^2\frac{N\Delta N(\mu+\Delta\mu, \lambda)}{mV^2}$. From this we see that $\Delta\mu \to 0$, since $\Delta N(\mu + \Delta\mu, \lambda)$ is of order one, in the thermodynamic limit where $N \to \infty$ and with $N/V$ fixed. Note that a bulk system can have a varying periodic potential, and so the results given here are not just valid for constant potentials.

For a generic trap, we assume that for large $\mu$ one has $N(\mu, 0) = (\frac{\mu}{\mu_0})^z$, where $\mu_0$ is an intrinsic energy scale, and as $N(\mu, 0)$ must increase with $\mu$ we must have $z > 0$. Indeed,

using the LDA in the bulk to compute $N(\mu, 0)$ as a function of $\mu$ for potentials of the form $V(x) \sim x^p$ we find

$$N(\mu, 0) = \int dx \rho_{0\mu}(x) = \frac{\sqrt{2m}}{\pi \hbar} \int_B dx \sqrt{\mu - V(x)} \,, \tag{146}$$

where $B$ denotes the bulk region where $\sqrt{\mu - V(x)}$ is real. Writing $V(x) = v|x|^p$ then gives

$$N(\mu, 0) = \frac{\sqrt{2m}}{\pi \hbar} \int_{-\left(\frac{\mu}{v}\right)^{\frac{1}{p}}}^{\left(\frac{\mu}{v}\right)^{\frac{1}{p}}} dx \sqrt{\mu - vx^p} = \frac{\sqrt{2m\mu}}{\pi \hbar} \left(\frac{\mu}{v}\right)^{\frac{1}{p}} \int_{-1}^{1} dy \sqrt{1 - y^p} \,, \tag{147}$$

so we find

$$N(\mu, 0) \simeq \left(\frac{\mu}{\mu_0}\right)^{\frac{1}{2} + \frac{1}{p}} \,, \tag{148}$$

and thus see that $z = \frac{1}{2} + \frac{1}{p}$. For $\mu$ large the condition in Eq. (145) reads

$$\frac{\Delta\mu}{\mu} = -\frac{\Delta N(\mu, \lambda)}{z N(\mu, 0)} = -\frac{\Delta N(\mu, \lambda)}{zN}, \tag{149}$$

and so we see that in the thermodynamic limit $\frac{\Delta\mu}{\mu} \to 0$. However

$$\Delta\mu = -\frac{\Delta N \mu_0}{z} N^{\frac{1}{z} - 1}, \tag{150}$$

and so only when $z > 1$ or, equivalently, when $p < 2$ we see that $\Delta\mu \to 0$.

In essence the results here are valid when the large energy states near the Fermi energy can be described as a continuum and the effects of discreteness can thus be neglected. Here a local analysis suffices to understand the physics. It would be interesting to extend the analysis to the cases where $\Delta\mu$ remains finite (for instance the case of the harmonic trap $p = 2$) or indeed diverges, traps with $p > 2$ and where, depending on the strength of the perturbation, the effects of discreteness in the spectrum of $H_0$ can be expected to play a role.

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
