# Peer review of "Impurities in systems of noninteracting trapped fermions"

_SciPost Physics_

## Round 1 · Referee Report · Anonymous (Referee 1) · 2021-2-10

Strengths

1- Well and carefully written paper. 2 – Detailed calculations. 3 – The filling transition is interesting, as well as the analogy to a phenomena in random matrix theory (BBP transition).

Weaknesses

1 – The study is a bit academic and too technical.
2 – This study does not add much to what was already known for the one-dimensional ideal Fermi gas (even in a trap and/or with delta impurities). The (huge) literature in solid-state physics and in atomic physics on non-interacting fermions in 1D is partly overlooked by the authors.

Report

In this theoretical work, the authors pursue the study of the one-dimensional non-interacting Fermi gas (this is the continuation of a series of at least seven papers). Their main focus is on the gas being trapped by an external potential and containing one or two delta-impurities that modify the density. At long distance, the impurities give rise to Friedel oscillations in the density, at short distance they deplete (repulsive) or enhance (attractive) the local density. When there are two impurities, they interact via a kind of Casimir interaction. There is an interesting filling transition when a sufficiently attractive impurity is placed outside of the support of the gas’ density.

The article is well and carefully written. The work is mainly technical and a bit academic considering the venerable age of the subject (almost a century, starting with Sommerfeld 1927). Do we learn something about non-interacting fermions that we did not know before? Maybe the filling transition. The rest, I believe, is mostly well-known, even if the exact formulas were maybe not obtained before. If it is really original, then the authors should show that they have carefully explored the literature. I suggest digging seriously in the following fields: solid-state physics (the electron gas., but also the XX spin ½ chain that, via Jordan-Wigner transformation, is equivalent to non-interacting fermions) and in cold atomic gases (either with spin-polarized fermionic atoms or with impenetrable bosonic atoms, the so-called Tonks-Girardeau gas). Examples of relevant papers: - on the inhomogeneous electron gas: Kohn and Sham, Phys. Rev. 137, A1697 (1965). See also Kohn and Majumdar, Phys. Rev. 138, A1617 (1965). - on the Tonks-Girardeau gas with an impurity (repulsive or attractive): Goold, Krych, Idziaszek, Fogarty and Busch, New J. Phys. 12, 093041 (2010). Fu and Rojo, PRA 74, 013620 (2006). But there are probably much more relevant references.

Questions/remarks: - in the title or in the abstract it is not mentioned that the study is restricted to one dimension. Although this becomes obvious in the core of the article, it should be specified in the title or at least in the abstract. - what is called the kernel in the present work is usually called the reduced single-particle density matrix in the solid-state literature on the Fermi gas (e.g. in Kohn and Majumdar, Phys Rev 1965). - the notation lambda introduced near equation (9) is a bit strange for a quantity which is the inverse of a length (usually lambda stands for a wavelength and k or q or kappa for an inverse length). In the same vein, the authors use k to label their eigenvalues and eigenvectors but here k is not the momentum which is not conserved due both to the potential and to the impurities. Nevertheless, they use k_F (which is the standard notation) to denote the Fermi momentum. This is a bit confusing. It would be better to have another label for quantum numbers k that are not momenta. - It would have been interesting to discuss the phase-shifts in the Friedel oscillations contained in equation (69). - Equation (82) is simply the standard mean-field result: interaction strength times local density = g rho.

Requested changes

1 - Authors should make clearer what was known before and what is truly new. This is especially important for a subject with such a long history. In particular, they should make a detailed comparison with Kohn and Sham 1965 and with results for delta-impurities in the Tonks-Girardeau gas.

2 - Experiments are mentioned in the conclusion but no reference is given. It would be interesting for the reader to have such an experimental motivation. For example, Paredes et al. Nature 2004; Wenz et al., Science 2013. But there are probably better references.

Typos:

3- in equation (70) a parenthesis stands alone in the denominator 4- just above section 7.2, “This is expected since that the edge...”: remove “that” 5- in footnote [33], fermi should be Fermi 6 - Acronyms (even well-known) should still be defined: GUE, PDF, RMT, etc.

---

## Round 1 · Referee Report · Anonymous (Referee 2) · 2021-2-16

Strengths

1-New results for an impurity in the vicinity of the 'Airy gas'.
2-Written in such a way that all technical steps are easy to reproduce.

Weaknesses

1-Relation to the broader existing literature.
2-Emphasis on non-universal physics.

Report

In this paper, the authors study non-interacting fermions in a potential, and the effects of one or several impurities on the density profile. They take a first principle microscopic approach based on Green's functions.

Overall the paper is a pleasant read, more on the technical side but with enough details to be able to reproduce the computations. The results on impurities near the edge are interesting and to my knowledge, new. The paper would be useful to a more mathematically minded audience with an interest in condensed matter physics. For this reason it deserves publication.

I'm less sure it would be that useful to a condensed matter theory audience, though. Most of the physical results presented in this paper would be considered well-known, especially for the free Fermi gas. While the authors correctly acknowledge the relation with Friedel oscillations, they do not do such a good job of explaining their results in a broader context. In particular, there is already a considerable literature on impurity effects even in interacting models, where some of what the authors get is modified (see for example the 1992 paper by Kane and Fisher).

The authors also insist several times that their short distance results have not been computed before, which is probably true. However, there is little hope to find universality in such short distance physics. The point of Friedel oscillations being that you do not really need an exact delta potential to observe those. Thus, they should make clear how much of what they are computing depends on the precise potential used to model the impurity.

I have other minor comments:

1) In figure 1, show also $V_{\rm eff}$.

2) Near (23), (24). Wouldn't it be simpler to say that the difference (23)-(24) is zero because there are no poles below the contour?

3) At the end of section 5. Since the joint PDF for random matrices is identical to the joint PDF for the fermions in a harmonic potential, what would be the fermionic analog of the BBP setup?

4) After (136). 'formula' should be plural.

Requested changes

1-Broader discussion of impurities in condensed matter physics.
2-Explain how much of this remains true for other impurity potentials.

---

## Editorial Decision

resubmitted